# Lipidomics reveals new lipid-based lung adenocarcinoma early diagnosis model

Ting Sun[1,6], Junge Chen [2,6], Fan Yang[3,4], Gang Zhang[5], Jiahao Chen[1], Xun Wang[3,4]✉ & Jing Zhang[1,2]✉

## Abstract

**Lung adenocarcinoma (LUAD) continues to pose a significant mortality risk with a lack of dependable biomarkers for early noninvasive cancer detection. Here, we find that aberrant lipid metabolism is significantly enriched in lung cancer cells. Further, we identified four signature lipids highly associated with LUAD and developed a lipid signature-based scoring model (LSRscore). Evaluation of LSRscore in a discovery cohort reveals a robust predictive capability for LUAD (AUC: 0.972), a result further validated in an independent cohort (AUC: 0.92). We highlight one lipid signature biomarker, PE(18:0/18:1), consistently exhibiting altered levels both in cancer tissue and in plasma of LUAD patients, demonstrating significant predictive power for early-stage LUAD. Transcriptome analysis reveals an association between increased PE(18:0/18:1) levels and dysregulated glycerophospholipid metabolism, which consistently displays strong prognostic value across two LUAD cohorts. The combined utility of LSRscore and PE(18:0/18:1) holds promise for early-stage diagnosis and prognosis of LUAD.**

**Keywords** LUAD; Lipidomics; Cancer Early Diagnosis Model; Lipid Metabolism; LSRscore
**Subject Categories** Biomarkers; Cancer; Respiratory System

## Introduction

Lung cancer remains the highest incidence of mortality among all tumors worldwide (Siegel et al, 2019; Siegel et al, 2020; Sung et al, 2021). Lung adenocarcinoma (LUAD) is the most common histological type of lung cancer. The 5-year overall survival (OS) rate of LUAD is ~18% (Fang et al, 2023) and the 5-year recurrence-free survival rate of stage IA ranges from 63 to 81% (Carr et al, 2023), although significant progress has been made in treating LUAD including surgical treatment, radiotherapy, chemotherapy,

target therapy and immune therapy. Thus, it is extremely meaningful and urgently needed for early cancer diagnosis to improve the overall survival of patients with LUAD. High-resolution computed tomography (CT) and low-dose computed tomography (LDCT) screening significantly improves the detection rate of early-stage lung cancer (Hoffman et al, 2020; International Early Lung Cancer Action Program I et al, 2006; National Lung Screening Trial Research T et al, 2011; van Klaveren et al, 2009), but high false-positive rate, radiation exposure, and high-cost limit their applications (Bach et al, 2012; Smith et al, 2019). Several blood-based tests have been developed to aid in early detection of lung cancer (Chen et al, 2019; Cohen et al, 2018; Shen et al, 2018), but their reliabilities and consistencies are inadequate and unsatisfactory. Therefore, the establishment of an effective and reliable way for early-stage LUAD diagnosis is still highly necessary.

Metabolome may reliably reflect the state of biological systems (Idle et al, 2007; Smolenska et al, 2015; Wang et al, 2022). As a subset of metabolomics, lipidomics is an effective approach to study the metabolism of the cellular lipidome and to identify lipid biomarkers of diseases (Han, 2016; Wenk, 2005). Plasma lipidomics has exhibited predictive powers in diagnosis of multiple diseases such as cancers (Lee et al, 2019), type 2 diabetes (Razquin et al, 2018), cardiovascular disease (Tabassum et al, 2019), and systemic lupus erythematosus (Chen et al, 2021). A recent research has reported the identification of nine lipids from plasma lipidomic profiling to be able to detect early-stage lung cancer (Wang et al, 2022), although their reliability remains to be confirmed from independent studies. As there is still a paucity of reliable methods for diagnosis of early-stage LUAD (Cohen et al, 2018; Shen et al, 2018; Wang et al, 2022), the identification of new lipid biomarkers and direct comparison of their performances with reported lipids would be critical to develop new reliable methods for early-stage diagnosis of LUAD.

Lipidomics has developed in an unprecedentedly fast speed to make clinical application soon possible (Meikle et al, 2014), which may be approached in an untargeted or targeted way. Data-dependent acquisition (DDA) for untargeted lipidomics has broad coverage of metabolites with wide applications in searching for biomarker candidates. Multiple reaction monitoring (MRM) mode-based targeted lipidomics has good sensitivity, accuracy, and

[1]Key Laboratory for Biomechanics and Mechanobiology of Ministry of Education, Beijing Advanced Innovation Center for Biomedical Engineering, School of Biological Science and Medical Engineering, Beihang University, 100083 Beijing, China. [2]Key Laboratory for Biomechanics and Mechanobiology of Ministry of Education, Beijing Advanced Innovation Center for Biomedical Engineering, School of Engineering Medicine, Beihang University, 100083 Beijing, China. [3]Department of Thoracic Surgery, Peking University People's Hospital, 100044 Beijing, China. [4]Thoracic Oncology Institute, Peking University People's Hospital, 100044 Beijing, China. [5]CAS Key Laboratory for Biomedical Effects of Nanomaterials and Nanosafety, CAS Center for Excellence in Nanoscience, National Center for Nanoscience and Technology of China, 100190 Beijing, China. [6]These authors contributed equally: Ting Sun, Junge Chen. ✉E-mail: wangxun04275@pkuph.edu.cn; jz2716@buaa.edu.cn

reliability (Ciccimaro et al, 2010; Kitteringham et al, 2009; Wei et al, 2010), which is widely used to validate biomarker candidates in independent sample cohorts. The combination of both untargeted and targeted lipidomics assays would significantly strengthen screening reliable power for LUAD-related lipids.

In this study, we explored the aberrant signal pathways in lung cancer cells through analyzing single-cell RNA sequencing (scRNA-seq) data of LUAD and healthy lung tissues, finding the altered lipid metabolism significantly enriched in tumor cells. By performing untargeted lipidomic profiling of lipids in plasma, we identified four signature lipids highly associated with LUAD. The plasma lipid signature-based ratio scoring model (LSRscore) was then developed and validated by targeted lipidomics assay in independent plasma cohorts of LUADs. Targeted lipidomics assay of early-stage LUAD cancer tissues in situ and adjacent normal lung tissues revealed that one signature lipid, PE(18:0/18:1), was consistently altered in both LUAD tissue in situ and plasmas. The concentration of PE(18:0/18:1) alone was found exhibiting strong predictive power for detecting early-stage LUADs. Transcriptome analysis independently confirmed that the overexpression of PE(18:0/18:1) was associated with the dysregulated glycerophospholipid metabolism, which was validated in vitro (dose-dependent). In addition, glycerophospholipid metabolism consistently demonstrated great prognostic power in LUAD cohorts. Thus, the signature lipid biomarker PE(18:0/18:1) and LSRscore had the potential to be used for early-stage diagnosis and predicting survival of LUAD.

# Results

## Aberrant lipid metabolism identified in tumor cells of early-stage LUAD

To discover reliable biomarkers for early-stage lung adenocarcinoma (LUAD) diagnosis, we collected single-cell RNA sequencing (scRNA-seq) data of 17 LUAD tumors and 17 LUAD adjacent normal and 8 healthy lung tissues from healthy donors in the public domain (Appendix Table S1). After removing low-quality cells, 170,435 cells were obtained. Harmony was used to remove batch effects across samples. We identified a total of seven major cell lineages based on well-established markers including B cells, endothelial cells, epithelial cells, fibroblasts, myeloid cells, cycling, and T and natural killer (NK) cells (Fig. 1A,B). The relative percentages of T cells and B cells increased and myeloid cells decreased in lung cancer tissues compared with those in healthy lung tissues (Fig. 1C), consistent with previous reports (Lambrechts et al, 2018; Lavin et al, 2017; Wang et al, 2022). Nonimmune cells, including endothelial cells, epithelial cells, fibroblasts, were retained for the downstream analysis (Appendix Fig. S1A,B). Lung tumor cells were separated from normal epithelia cells by large-scale segmented copy number variations (CNVs) predicted from expression profiles using inferCNV (Appendix Fig. S1C). Malignant cells, usually having a high degree of aneuploidy formed separate clusters from nonmalignant cells independent of their sample of origin (Fig. 1D,E). Differential gene expression analysis between malignant and nonmalignant epithelial cells revealed that down-regulated genes were enriched in lipid metabolism-related biological processes, including lipid metabolism and localization,

phospholipid metabolism, and fatty acid metabolism (Fig. 1F; Appendix Table S2). These results demonstrated that lipid metabolism dysregulation occurred in early-stage lung cancer, highly consistent with previous study (Wang et al, 2022). We hypothesized that aberrant lipids might be useful in diagnosing early-stage LUAD.

## Clinical characteristics

To identify aberrant lipid signature, we collected plasma and tissue samples. Clinical characteristics of LUAD patients and healthy subjects in the discovery cohort (plasma collected for untargeted lipidomic assay), plasma validation cohort (plasma collected for targeted lipidomic assay) and tissue validation cohort (tumor and adjacent normal tissues collected for targeted lipidomic assay) were collected from Peking University People's Hospital and presented in Appendix Tables S3–5. The distribution of tumor grade was similar between the discovery cohort and the plasma validation cohorts, and most patients were stage I LUAD. The age and gender of patients (age: $59.2 \pm 10.73$ years, female: $n = 32$, male: $n = 28$) and healthy subjects (age: $29.73 \pm 5.99$ years, female: $n = 7$, male: $n = 23$) was significantly different in the discovery cohort. The age and gender of patients (age: $62.23 \pm 9.48$ years, female: $n = 15$, male: $n = 15$) and healthy controls (age: $51.57 \pm 4.52$ years, female: $n = 10$, male: $n = 20$) were similar in the plasma validation cohort. The average age of 25 LUADs with tumor and adjacent lung tissues available was 62.72 years old (age: $62.72 \pm 10.21$ years, female: $n = 11$, male: $n = 14$).

## Untargeted lipidomics profiling of plasmas from LUADs

To comprehensively and unambiguously identify and quantify the expression of abnormal lipid metabolism in LUADs, we collected preoperative plasma samples from 60 LUADs (50 stage I, 7 stage II, and 3 stage III/IV) and plasma samples from 30 healthy subjects used as the healthy control group (HC). Untargeted lipidomics profiling were performed on plasma from both LUADs and HCs using ultrahigh-performance liquid chromatography coupled with Q-Exactive MS/MS (UHPLC-QE-MS).

MS/MS spectra on data-dependent acquisition (DDA) mode were acquired and processed for retention time correction, peak identification, peak extraction, peak integration, and peak alignment ("Methods"). A total of 385 lipids of 9 lipid classes and 272 lipids of 12 lipid classes were identified in positive and negative ion modes, respectively.

## Identification of lipid signature for LUAD from untargeted plasma lipidomics data

To investigate whether LUADs underwent significant lipid change patterns compared with HC, we created orthogonal partial least squares discriminant analysis (OPLS-DA) models from all lipids detected in the entire discovery set at positive or negative ion mode, respectively. Clear separations were observed between LUAD and HC (Fig. 2A for positive mode, Fig. 2B for negative mode). Cross-validation of OPLS-DA models obtained from 200 permutation tests estimated Q2-intercept values to be −0.57 and −0.491, and R2-intercept values to 0.69 and 0.534 at positive or negative ion mode, respectively, confirming that OPLS-DA models had no overfitting and was credible (Appendix Fig. S2A,B). These results

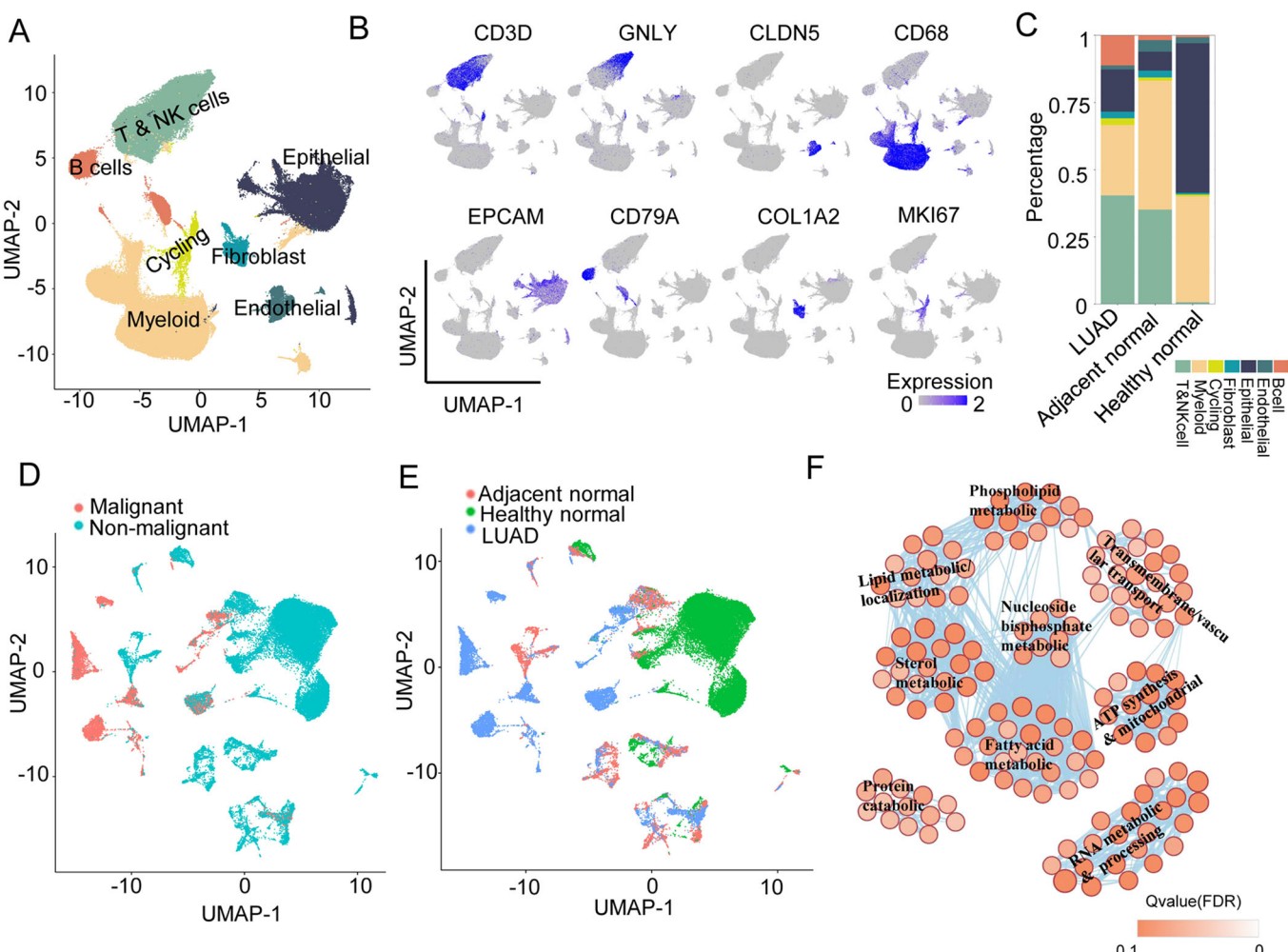

**Figure 1.  scRNA-seq analyses of early-stage LUAD and healthy lung tissues identifies aberrant lipid metabolism in cancer cells.**

(A) The uniform manifold approximation and projection (UMAP) of 170435 single cells identified by scRNA-seq in tumor and adjacent normal tissues from 17 LUAD patients and healthy lung tissues from eight healthy donors. Seven major cell subpopulations are labeled. Each dot is a single cell and colored based on its cell cypt. (B) Canonical markers of different cell types used to identify cell subpopulations in UMAP plot. (C) The distribution of cell subpopulations in LUAD tumor tissues, adjacent normal tissues and healthy lung tissues. Each color represents a cell subpopulation. (D) Lung tumor cells and normal epithelial and stromal cell subpopulations. Red is malignant cells; blue is nonmalignant cells. (E) The sample of origin for malignant and nonmalignant cells. Red is adjacent normal lung tissues; green is healthy normal lung tissues; blue is LUAD tumor tissues. (F) Enrichment map network of statistically significant GO categories in the malignant cells. Nodes are GO categories and lines their connectivity. Node size is proportional to the number of genes in the GO term and line thickness is the fraction of genes overlapped between GO categories.

indicated that there were noticeable variations in lipid metabolism between LUAD and HC, suggesting that lipids profiling could be used for diagnosis of LUAD.

We next applied Mann–Whitney $U$ test upon the normalized untargeted lipid data and identified a total of 149 differential lipids (FDR ≤ 0.05; Fold change >1.5 or <1/1.5; VIP (Variable importance in the projection, OPLS-DA) > 1) in plasma of LUADs compared with that of HCs, including 50 upregulated and 48 downregulated lipids in positive ion mode, 39 upregulated and 12 downregulated lipids in negative ion mode (Fig. 2C,D). To determine lipid signature for LUAD, we performed 2000 iterations of random sampling on the untargeted lipid data of differential lipids, in which 90 samples were randomly divided into a training set ($n = 63$, 70% of all samples) and a test set ($n = 27$, 30% of all samples) (Appendix

Fig. S3A). Random Forest and XGBoost were independently applied to classify LUADs versus HCs. A third method, Lasso regression, was also used upon all data (90 samples) of differential lipids to infer lipid signature candidates associated with LUAD. The untargeted lipid data of differential lipids in positive and negative ion modes were analyzed separately. To test the robustness of lipid signature selection for LUAD, we randomly split the samples in the discovery cohort into a training set and a test set at different ratios of 6:4, 7:3, 8:2, 9:1, and 10:0, respectively. We discovered that the features selected at different scenarios were highly consistent (Appendix Fig. S3B). We obtained a total of seven differential lipids (three lipids detected in positive ion mode, and 4 lipids detected in negative ion mode) that were consistently inferred by all three methods. Hierarchical clustering demonstrated that the lipid

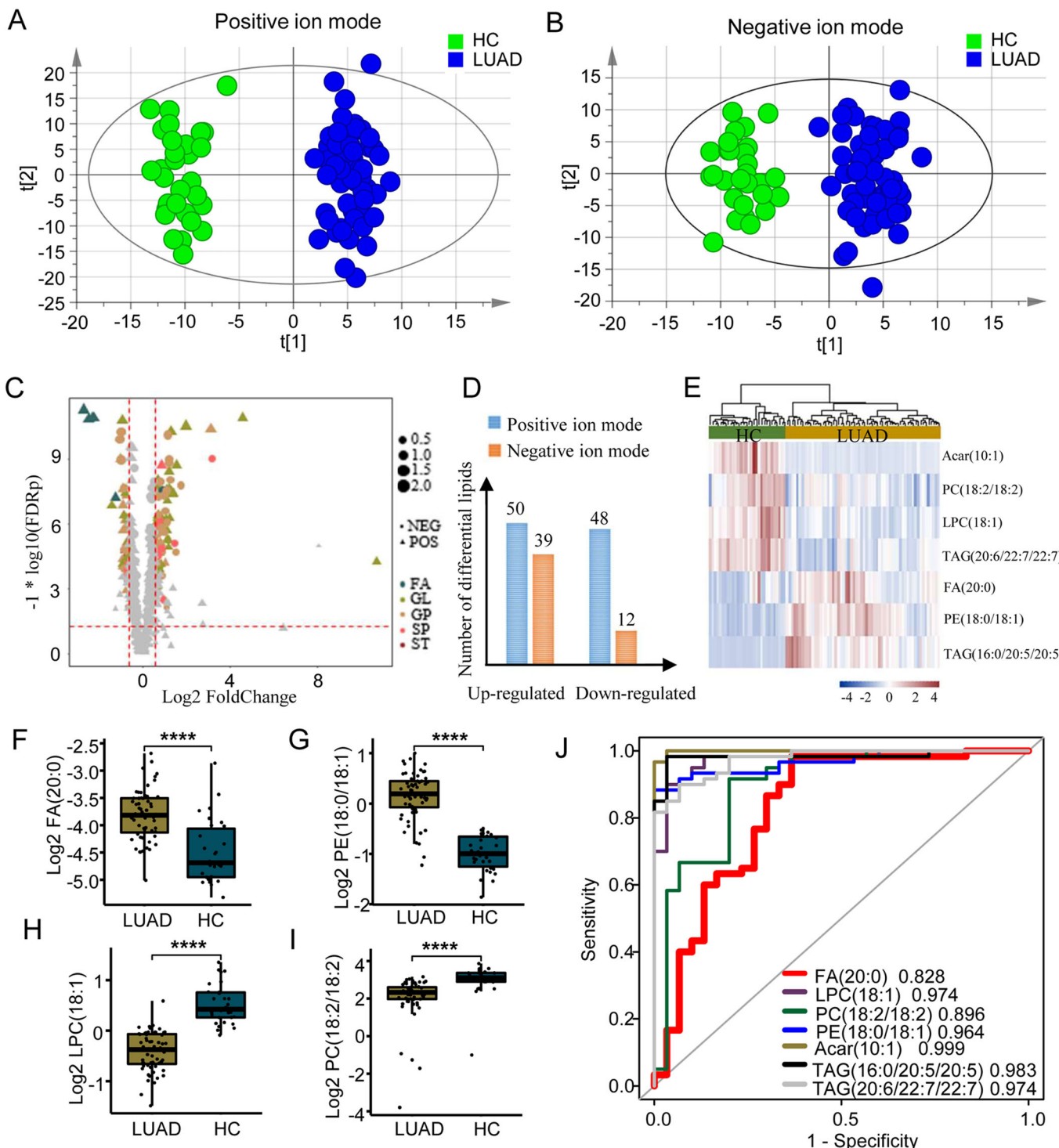

signature achieved 100% accuracy of classifying LUADs from HCs (Fig. 2E).

## Construction of LSRscore, a lipid signature-based predictive model for LUAD

As accurate detection and quantification of lipid abundance are required in actual clinical application, it is essential to validate the

predictive power of the lipid signature through LC–MS-based targeted assay which requires internal standards. We found there were internal standards available for four of the seven signature lipids with FA(20:0) (Fig. 2F) and PE(18:0/18:1) (Fig. 2G) being upregulated, and LPC(18:1) (Fig. 2H) and PC(18:2/18:2) (Fig. 2I) being downregulated in LUADs compared with HCs. We then evaluated the capability of each lipid in the signature distinguishing LUADs from HCs through receiver operating characteristics (ROC)

**Figure 2.   The signature lipids for early-stage LUAD detection, n = 60 for LUAD and 30 for HC.**

(A, B) Score plot of OPLS-DA model for LUAD and HC on the discovery cohort at positive ion mode (**A**) and at negative ion mode (**B**), respectively. Each dot represents a sample. Green is healthy subjects used as control (HC); blue is LUAD patients. (**C**) Differential lipids identified at positive or negative ion model, respectively. X axis is a base-2 logarithmic scale of fold change for lipid. Y axis is the negative of the base-10 logarithmic scale of FDR value. Triangle represents positive ion mode. Dot represents negative ion mode. Size of each triangle or dot indicates variable importance in the projection (VIP) values. Differential lipids are colored according to lipid class. FA fatty acids, GL glycerolipids, GP glycerophospholipids, SP sphingolipids, ST sterols. (**D**) The number of differential lipids identified at positive or negative ion mode, respectively. (**E**) The hierarchical clustering of samples using the 7 signature lipids for LUAD. The clustering method was used ward D2. (**F–I**) The comparison of lipid concentration between HC and LUAD for FA(20:0) (**F**), PE(18:0/18:1) (**G**), LPC(18:1) (**H**), and PC(18:2/18:2) (**I**). **** represents P value from the Mann–Whitney U test <0.0001. For box, bottom boundary presented as 25% quantile, upper boundary presented as 75% quantile and line in the middle presented as median value of data. Upper whisker: largest observation less than or equal to 75% quantile + 1.5 *iqr; lower whisker: smallest observation greater than or equal to 25% quantile − 1.5 *iqr;iqr = 75% quantile − 25% quantile. (**J**) ROC curves of signature lipids on the discovery cohort.

analysis, discovering that all signature lipids achieved high values of area under the curve (AUC) varying from 0.828 to 0.999 (Fig. 2J). These data demonstrated that the four signature lipids were reliable and suitable for targeted quantification analysis, having large potentials to diagnose LUADs.

Inspired by the observations that lipid ratios exhibited good predictive power for multiple diseases (Hadaegh et al, 2010; Yang et al, 2020; Yu et al, 2021), we constructed a lipid signature-based scoring model which scored each sample by the sum of the concentration of two upregulated signature lipids (FA(20:0) and PE(18:0/18:1)) divided by the sum of the concentration of two downregulated ones (LPC(18:1) and PC(18:2/18:2)). We named this lipid signature-based ratio scoring model as LSRscore. ROC analysis of untargeted lipidomics data demonstrated that LSRscore achieved good performance with AUC equal to 0.972 (Fig. 3A). Compared with each individual signature lipid included in LSRscore, LSRscore achieved similar AUC value as LPC (18:1) (AUC = 0.974), but higher than FA(20:0) (AUC = 0.828), PC(18:2/18:2) (AUC = 0.896), and PE(18:0/18:1) (AUC = 0.964). Multivariate binary logistic analysis also showed that LSRscore was an independent factor predictive of LUAD after adjusting for other clinical factors (Fig. 3B).

## Independent validation of the performance of LSRscore in predicting LUAD

To examine the performance of LSRscore on an independent cohort, we newly collected plasmas from 30 LUAD patients (diagnosed with lung adenocarcinoma including 29 TNM stages I to II and 1 stage III) and 30 healthy participants from Peking University People's Hospital. An LC–MS-based targeted lipidomics assay using multiple reaction monitoring (MRM) were then applied to detect the concentrations of the four lipids, which were then used to calculate LSRscore for each sample. ROC analysis revealed that LSRscore achieved much better predictive power for LUAD with a high AUC value of 0.92 than each individual lipid which had varying AUC values from 0.62 to 0.84 (AUC: 0.62 for FFA(20:0), 0.65 for LPC(18:1), 0.84 for PC(18:2/18:2), and 0.80 for PE(18:0/18:1)) (Fig. 3C). Multivariate binary logistic analysis also showed that LSRscore was a predictive factor for LUAD independent of other clinical factors including age and gender (Fig. 3D). These results demonstrated the high accuracy and specificity of LSRscore in LUAD diagnosis.

We next examined the levels of the four lipids in plasma from LUADs and HCs, respectively. The levels of PE(18:0/18:1) (Fig. 3F) were found to statistically significantly increase (P < 0.0001), and that of PC(18:2/18:2) (Fig. 3H) significantly decreased (P < 0.0001) in plasmas from LUADs compared with HCs, highly consistent with the findings in the discovery set. The levels of LPC(18:1) (Fig. 3G) were confirmed lower in plasmas of LUADs than HCs but only reaching marginal significance (P = 0.053). FA(20:0) exhibited no difference between LUAD and HCs (Fig. 3E) (P = 0.10). These data suggested that PE(18:0/18:1) and PC(18:2/18:2) could be critical players in classifying LUADs from HCs.

## The increased levels of PE(18:0/18:1) highly consistent in LUAD tissues and plasmas

To further investigate the alterations of PE(18:0/18:1) and PC(18:2/18:2) in cancer tissues of LUADs, we measured their concentrations in LUAD cancer tissues and adjacent lung tissues of 25 LUAD patients through LC–MS-based targeted lipidomics assay using MRM. Both PE(18:0/18:1) and PC(18:2/18:2) were detected in these surgically resected tissues. Compared with adjacent lung tissues, PE(18:0/18:1) was significantly upregulated in LUAD tumor tissues (Mann–Whitney U test:P = 1.1e–5; Paired t test: P = 2.5e–5) (Fig. 4A,B). But there were no significant alterations observed for PC(18:2/18:2) between LUAD tumor tissues and adjacent lung tissues (Mann–Whitney U test: P = 0.1; Paired t test: P = 0.06) (Fig. 4C,D). ROC analysis revealed that PE(18:0/18:1) achieved good performance with AUC value reaching 0.845 (Fig. 4F). PC(18:2/18:2) had an AUC value of 0.637 (Fig. 4E). These results confirmed that PE(18:0/18:1) were consistently altered in LUAD tissues. Altogether, multiple layers of independent evidence from plasma and tissues in situ demonstrated that PE(18:0/18:1) alone had strong predictive power for diagnosing early-stage LUAD (AUC = 0.964 in untargeted lipidomics of plasma cohort, Fig. 3A; AUC = 0.80 for targeted lipidomics of independent plasma cohort, Fig. 3C; AUC = 0.845 for targeted lipidomics of LUAD tissues in situ, Fig. 4F).

To validate that PE(18:0/18:1) is specifically highly expressed in LUAD, we applied targeted lipidomics analysis upon two LUAD cell lines (A549 and NCI-H1975), and three control cell lines, including two normal human cell lines (human normal lung epithelial cells, BEAS-2B; human skin fibroblasts, HSF) and one human liver carcinoma cell line (HepG2), with each having three replicates (Appendix Table S6). We discovered that the levels of PE(18:0/18:1) were significantly higher in LUAD cells than that in normal control and liver carcinoma cell lines (t test, P = 0.004 for A549 and BEAS-2B; P = 0.05 for A549 and HSF; P = 0.01 for NCI-H1975 and BEAS-2B; P = 0.026 for NCI-H1975 and HSF; P = 0.015 for A549 and HepG2; P = 0.02 for NCI-H1975 and HepG2)

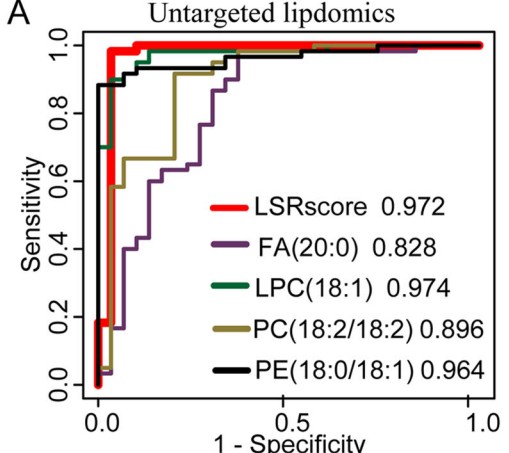

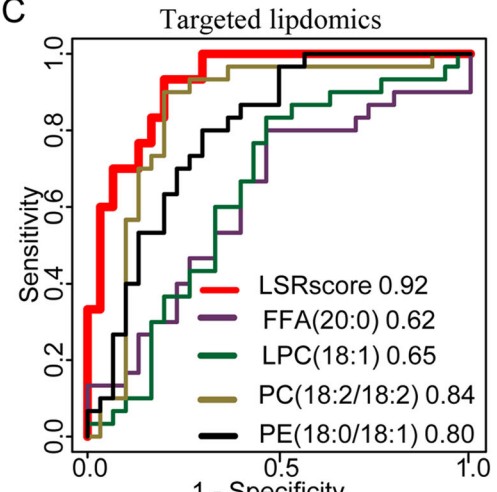

**A** Untargeted lipdomics

LSRscore 0.972
FA(20:0) 0.828
LPC(18:1) 0.974
PC(18:2/18:2) 0.896
PE(18:0/18:1) 0.964

**B** Multivariate logistic analysis in untargeted lipdomics data

| Variables | Wald | OR(95% CI) | P value |
|---|---|---|---|
| LSRscore | 11.031 | 1.429(1.157~1.763) | 0.0013 |
| Age | 143.186 | 1.021(1.017~1.024) | < 0.001 |
| Gender | 7.509 | 1.152(1.041~1.275) | 0.007 |
| TumorMarker | 2.512 | 1.104(0.977~1.248) | 0.117 |

**C** Targeted lipdomics

LSRscore 0.92
FFA(20:0) 0.62
LPC(18:1) 0.65
PC(18:2/18:2) 0.84
PE(18:0/18:1) 0.80

**D** Multivariate logistic analysis in targeted lipdomics data

| Variables | Wald | OR(95% CI) | P value |
|---|---|---|---|
| LSRscore | 11.857 | 6.108(2.605~21.806) | 0.0006 |
| Age | 6.339 | 1.228(1.083~1.513) | 0.0118 |
| Gender | 0.137 | 1.474(0.203~14.606) | 0.7112 |

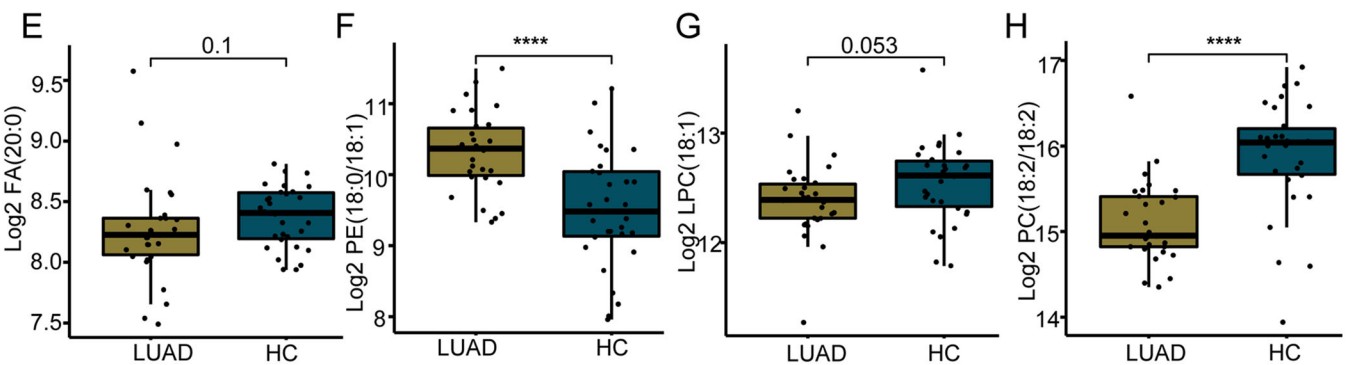

**Figure 3. The classification performance of LSRscore on the discovery and validation cohorts.**

(A) ROC curves of LSRscore and the four involving signature lipids on the discovery cohort ($n = 90$). (B) The multivariate logistic analysis of LSRscore and clinical factors including age, gender, and tumorMarker ($n = 90$). (C) ROC curves of LSRscore and the four involving signature lipids on the prospective validation cohort. (D) The multivariate logistic analysis of 100*LSRscore and clinical factors including age, and gender. (E–H) The comparison of lipid concentration between HC and LUAD for FA(20:0) (E), PE(18:0/18:1) (F), LPC(18:1) (G), and PC(18:2/18:2) (H). **** represents P value from Mann–Whitney U test <0.0001. For box, bottom boundary presented as 25% quantile, upper boundary presented as 75% quantile and line in the middle presented as median value of data. Upper whisker: largest observation less than or equal to 75% quantile + 1.5 *iqr; lower whisker: smallest observation greater than or equal to 25% quantile − 1.5 *iqr;iqr = 75% quantile − 25% quantile. Data information: (C–H) $n = 30$ samples for each group.

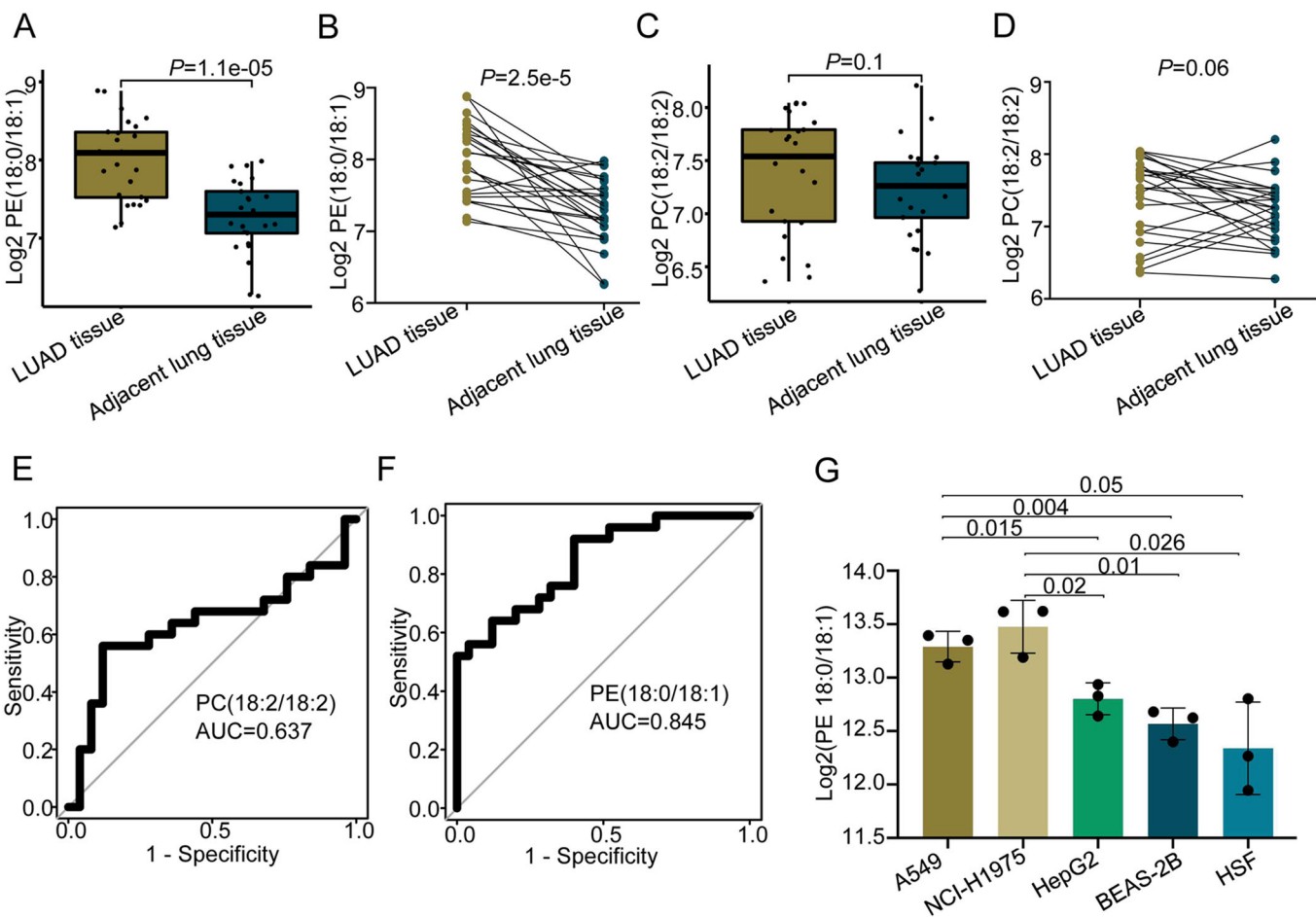

**Figure 4. The classification performance of PE(18:0/18:1) on tissue validation cohort, $n = 25$ samples for each group.**

(A, B) The comparison of lipid concentration between LUAD cancer tissue and adjacent lung tissue using Mann–Whitney $U$ test (A) and paired Student $t$ test (B) for PE(18:0/18:1). (C, D) The comparison of PC(18:2/18:2) concentration between LUAD cancer tissue and adjacent lung tissue using Mann–Whitney $U$ test (C) and paired Student $t$ (D), respectively. (E, F) ROC curves of PC(18:2/18:2) (E) and PE(18:0/18:1) (F). (G) The comparison of PE(18:0/18:1) concentration between LUAD cell lines (A549,NCI-H1975) and control cell lines (HepG2,BEAS-2B,HSF) (each having 3 replicates) using Student $t$ test. Data information: (A, C) For box, bottom boundary presented as 25% quantile, upper boundary presented as 75% quantile and line in the middle presented as median value of data. upper whisker: largest observation less than or equal to 75% quantile + 1.5 *iqr; lower whisker: smallest observation greater than or equal to 25% quantile − 1.5 *iqr;iqr = 75% quantile − 25% quantile.

(Fig. 4G). In addition, the level of PE(18:0/18:1) in liver carcinoma cell line was not significantly higher than those in normal cell lines ($t$ test, $P = 0.13$ for HepG2 and BEAS-2B; $P = 0.2$ for HepG2 and HSF) (Fig. 4G). There was no significant difference either between two LUAD cell lines ($t$ test, $P = 0.34$ for A549 and NCI-H1975) or between two normal cell lines ($t$ test, $P = 0.46$ for BEAS-2B and HSF) (Fig. 4G). These results from in vitro confirmed that PE(18:0/18:1) is specifically upregulated in LUAD.

## Aberrant alteration of pathways associated with PE(18:0/18:1) in tumor tissues of early-stage LUAD

To explore the pathways that could be influenced by PE(18:0/18:1), we collected RNA sequencing data of 487 primary tumor from TCGA-LUAD (Data ref: Cancer Genome Atlas Research N 2014), 229 subjects from CPTAC-LUAD and 120 healthy lung tissues from the public database (Data ref: Lee et al, 2021; Data ref: Sivakumar et al, 2019; Data ref: Suntsova et al, 2019). A direct

comparison of tumor and normal lung tissues revealed that six of seven KEGG pathways incorporating PE(18:0/18:1) (Appendix Table S7) showed significant aberrant alterations as demonstrated by gene set variation analysis (Fig. 5A). Particularly, glycerophospholipid metabolism and retrograde endocannabinoid signaling were downregulated, and glycosylphosphatidylinositol (GPI)-anchor biosynthesis was upregulated in LUAD cancer tissues compared with normal lung tissues (Fig. 5A; Appendix Fig. S4A), suggesting that the overexpression of PE(18:8/18:1) could make important contribution to the disruption of the lipid metabolisms in early stage of LUAD.

To examine whether disrupted lipid metabolisms may influence the survival of LUADs, we calculated Normalized Enrichment Score (NES) for each metabolic pathway and found LUADs highly enriched with glycerophospholipid metabolism had significantly better survival as demonstrated by Kaplan–Meier analysis in both TCGA-LUAD (log-rank $P = 0.029$, Appendix Fig. S4B; Appendix Table S8) and CPTAC-LUAD (log-rank $P = 0.00014$, Fig. 5B;

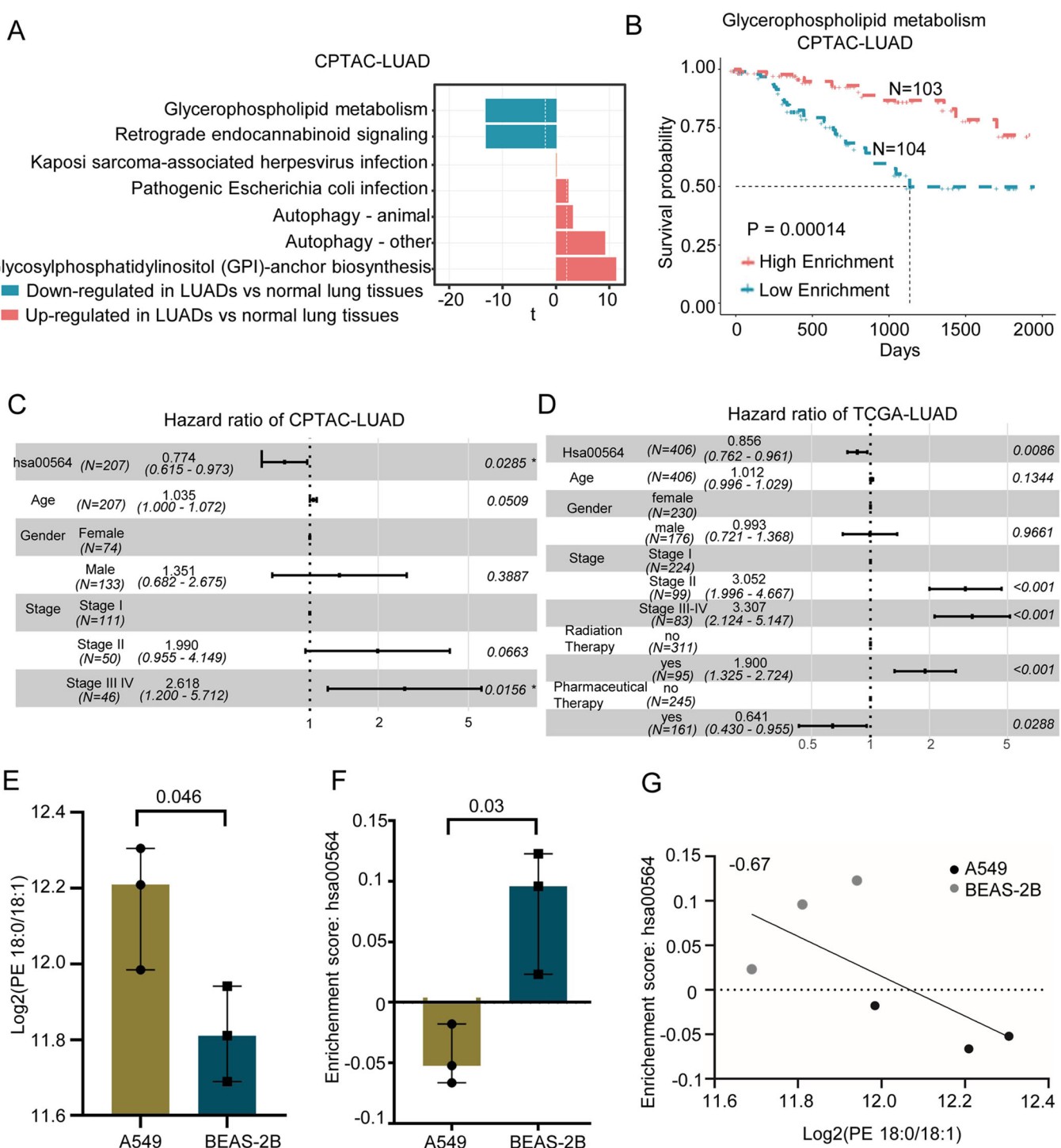

**Figure 5. The prognostic values of KEGG pathways associated with PE(18:0/18:1).**

(A) The enrichment status of KEGG pathways involving PE(18:0/18:1) in CPTAC-LUAD. n = 229 LUADs and 120 Healthy lung tissues. (B) Kaplan–Meier curves show the overall survival of LUAD patients with high (> median) or low (≤ median) enrichment of glycerophospholipid metabolism in CPTAC-LUAD. P value was calculated by log-rank test. (C, D) Multivariate cox regression analysis reveal that glycerophospholipid metabolism is associated with overall survival independent of other clinical factors on CPTCA-LUAD (C) and TCGA-LUAD cohorts (D), respectively. Line segment represents the 95% confidence interval of hazard ratio (HR), while points represent the HR value calculated by cox regression. (E, F) The level of lipid PE(18:0/18:1) (E) and the enrichment status of glycerophospholipid metabolism pathways (F) between LUAD A549cell line and healthy BEAS-2B cell line with three independent repetitions. Welch's t test P value less than 0.05 was considered significant. The error bar means the min and max value of data, while the bar is the median value. (G) The Pearson correlation of enrichment scores of glycerophospholipid metabolic pathways with PE(18:0/18:1) levels across A549 and BEAS-2B cell lines. The experiment was independently repeated three times.

Appendix Table S9). Multivariate Cox regression analysis revealed that glycerophospholipid metabolism was an independent factor predictive of better survival of LUADs after adjusting for other clinical factors, including age, gender, tumor stage, and treatment information (available for LUADs in TCGA) in both CPTAC-LUAD (Fig. 5C) and TCGA-LUAD(Fig. 5D). The enrichment scores of metabolic pathways did not exhibit significantly different between patients with and without treatments in TCGA-LUAD (Appendix Fig. S4C). In conclusion, PE(18:0/18:1) could help detect early-stage LUADs. Dysregulated glycerophospholipid metabolism involved by PE(18:0/18:1), was able to predict the survival of LUAD patients as validated in two independent large LUAD cohorts.

To further validate the association between PE(18:0/18:1) and dysregulated glycerophospholipid metabolism, we performed targeted lipidomics assay and RNA sequencing upon cultured LUAD cells(A549) and human normal lung epithelial cells (BEAS-2B) with each having three replicates. The levels of PE(18:0/18:1) were significantly higher in LUAD cells than in healthy controls ($t$ test, $P = 0.046$, Fig. 5E; Appendix Table S10). We then calculated the enrichment score of glycerophospholid metabolism using RNA-seq data of A549 and BEAS-2B, finding that glycerophospholid metabolism had a significantly lower enrichment score in A549 cells than in BEAS-2B cells ($t$ test, $P = 0.03$, Fig. 5F; Appendix Table S10), confirming our discovery that the overexpression of PE(18:0/18:1) is associated with the dysregulated glycerophospholipid metabolism (Fig. 5G).

To verify whether the levels of PE(18:0/18:1) may directly affect the dysregulation of glycerophospholipid metabolism, we treated BEAS-2B cell lines with PE(18:0/18:1) applied at different concentrations (0 μmol/L, 1 μmol/L, 10 μmol/L, and 100 μmol/L), followed by RNA sequencing (each having three replicates). GO enrichment analysis demonstrated that there were no significantly enriched GO terms between BEAS-2B cell lines treated with PE(18:0/18:1) at 1 μmol/L and those at 0 μmol/L (as control). When the PE(18:0/18:1) concentration increased to 10 μmol/L, we observed that some lipid metabolism-related pathways were downregulated in BEAS-2B cell lines (Appendix Fig. S5A,B). Particularly, a great number of lipid metabolism-related pathways, including glycerophospholipids metabolism were downregulated in BEAS-2B cell lines treated with PE(18:0/18:1) at 100 μmol/L (Appendix Fig. S5C–E). There was a clear trend that the increasing concentration of PE(18:0/18:1) was associated with the decreasing enrichment score of glycerophospholipid metabolism (Appendix Fig. S5F).

## Discussion

Early-stage LUAD diagnosis model could be helpful in identifying patients who may benefit from early clinical interventions. In this study, we identified four signature lipid biomarkers significantly altered in plasmas of LUADs through performing untargeted lipidomics profiling and developed a plasma lipid signature-based ratio scoring model (LSRscore) to effectively diagnose early-stage LUADs with AUC value reaching 0.972 in the discovery cohort. MRM-based targeted lipidomics combined with internal standards and a standard curve is viewed as the gold standard, making the quantification of target lipids more accurate and reliable (Cicci-maro et al, 2010; Kitteringham et al, 2009; Wei et al, 2010).

Targeted assay was applied to evaluate the performance of LSRscore for early-stage LUAD diagnosis in an independent cohort of plasmas from LUAD patients. LSRscore achieved a high AUC value of 0.924, exhibiting its strong predictive power in diagnosing early-stage LUAD.

The high accuracy and specificity of LSRscore manifest its huge potential in clinical applications to improve clinical detection and screening for LUAD without invasive diagnostic procedures or unnecessary exposure to radiation as required by LDCT which has a low positive predictive value ranging from 4 to 40% (National Lung Screening Trial Research T et al, 2011). ctDNA is also a noninvasive diagnostic method but with a short half-life (<2.5 h), and moreover, low peripheral blood ctDNA levels may result in MAFs below the detection limit of existing cancer early detection technologies, leading to false-negative results in clinical testing (de Rubis et al, 2019). Compared with ctDNA, targeted lipidomics was a more stable method. The combination of targeted lipidomics assay and LSRscore may be another useful way to detect early LUAD in addition to LDCT screening and current diagnostic procedures.

The four lipids in LSRscore including FA(20:0), LPC(18:1), PC(18:2/18:2), and PE(18:0/18:1) had varying degree of predictive power individually, which were consistently confirmed in both discovery and validation cohort of plasmas from LUADs. The distribution of the levels of the four lipid markers in plasma also varied between patients, indicating the characteristics of hetero-geneity among LUADs in regards to lipid metabolism, consistent with previous reports (Hensley et al, 2016) (Appendix Fig. S6). We also noticed that the levels of lipid markers and LSRscore were not affected by age (Appendix Fig. S7), smoking status, EGFR mutations, COPD (chronic obstructive pulmonary disease) and bronchiectasis (Appendix Fig. S8), confirming the stability of lipid marker and LSRscore classification. Particularly, PE(18:0/18:1) constantly exhibited powerful capability in detecting early-stage LUAD. PE(18:0/18:1) is physiologically involved in glycerophospholipid metabolism. Two independent transcriptome analyses of LUAD tumor tissues and normal lung tissues revealed that glycerophospholipid metabolism was deregulated in tumor tissues from early-stage LUAD. scRNA-seq analyses also demonstrated the significant enrichment of altered lipid metabolism in LUAD tumor cells. Moreover, PE(18:0/18:1) was found consistently altered in both LUAD cancer tissue in situ and plasma by targeted lipidomics assay of cancer tissues in situ and adjacent normal lung tissues from 25 early-stage LUAD patients. Therefore, we hypothesize that the overexpression of PE(18:0/18:1) detected in plasma could be associated with dysregulated lipid metabolism in LUAD tumor tissues. Interestingly, the levels of PE(36:1), the synonym of PE(18:0/18:1), was also discovered significantly different between pancreatic ductal adenocarcinoma (PDAC) and normal control in one cohort (Wolrab et al, 2022). It still needs further validation about the performance of PE(36:1) in diagnosing PDAC. In addition, the levels of PE(36:1) in tumor tissues of PDAC are also needed to be extensively evaluated. In addition, we examined the levels of PE(18:0/18:1) in two LUAD cell lines, two normal human tissue cell lines and one liver carcinoma cell line, confirming that PE(18:0/18:1) specifically significantly over-expressed in LUAD cells. Dose-dependent effects of PE(18:0/18:1) on BEAS-2B demonstrated that PE(18:0/18:1) influenced glycerophospholipid metabolism.

A very recent study reported an LC–MS-based targeted lipidomics assay using MRM model (LCAID v2.0), which included nine lipids of PC, TG, and LPC (Wang et al, 2022). We evaluated whether the nine lipids in LCAID v2.0 had predictive power for LUAD in our untargeted lipidomics data. We discovered that only four (LPC(16:0), LPC(20:4), PC(16:0/22:6), and PC(18:0/18:1)) of the nine lipids were detected in the plasma of our LUAD cohort, which could significantly limit the application of LCAID v2.0 in early cancer detection. In our analysis of searching for lipid markers, their four lipid markers were filtered due to their fold changes smaller than 1.5 (or larger than 0.67) between LUADs and healthy people in the Discovery cohort (PC 16:0–22:6 and PC 18:0–18:1 being upregulated with fold changes of 1.34 and 1.44, LPC 16:0, and LPC 20:4 being downregulated with fold change of 0.77 and 0.71 in LUAD compared with healthy people). We then compared the performance of their four lipid markers with that of our four lipid markers in the Discovery cohort, finding that their four lipid markers performed worse than our four lipid signature (Appendix Fig. S9). In contrast, our LSRscore model or PE(18:0/18:1) may be a better choice in early-stage LUAD diagnosis and have wider applications. Further investigations from independent sources are needed to evaluate and benchmark the performance of LSRscore model and other lipid-based models such as LCAID v2.0.

Limitations of the present study should also be recognized. First, the LSRscore model could be independently validated on a larger independent sample set, and further comparisons of the PE (18:0/18:1) levels in a wider range of tumor types should be performed. Lipid metabolism among different lung cancer types such as squamous carcinoma (LUSC) (predominantly early-stage lung adenocarcinoma in our cohort) can be further investigated.

In summary, we developed and validated a newly identified lipid signature-based model, LSRscore, which achieved good performance for early-stage LUAD diagnosis in independent cohorts of plasmas from LUADs. Targeted lipidomics assay of early-stage LUAD cancer tissues in situ and adjacent normal lung tissues showed that PE(18:0/18:1) incorporated in LSRscore was consistently altered in both LUAD tissue in situ and plasmas. Moreover, the concentration of PE(18:0/18:1) alone exhibited strong predictive power for detecting early-stage LUADs. scRNA-seq and transcriptome analyses independently confirmed that the overexpression of PE(18:0/18:1) was associated with dysregulated lipid metabolism. This study demonstrated that LSRscore and PE(18:0/18:1) had large potential in the clinical application of early-stage diagnosis and predicting survival of LUAD.

# Methods

## Patient enrollment

All human sample collection and usage were approved by the institution review board of Peking University People's Hospital (Ethical Approval No. 2020PHB220-02) and were conducted according to the principles of the Declaration of Helsinki and the Department of Health and Human Services Belmont Report. Informed consent was obtained from the enrolled participants. Three independent cohorts of a total of ~150 subjects with plasma samples and 25 subjects with tumor tissues in situ and adjacent normal lung tissues, and clinicopathological characteristics were included in this study. For the discovery cohort, plasma samples of 60 patients from at Peking University People's Hospital were collected retrospectively according to the following criteria: pathologically confirmed LUADs, no history of other cancers, over 18 years old, no chronic blood system diseases (such as hemolytic diseases), and no anticancer treatment such as chemotherapy, radiotherapy, targeted therapy and immune therapy. In addition, the discovery cohort also included plasma samples of 30 healthy subjects.

In the prospective validation cohort (plasma), plasma samples of 30 patients with LUAD were consecutively enrolled in this study from June 2020 to September 2021 at Peking University People's Hospital. And plasma samples of 30 healthy subjects were collected from Peking University People's Hospital.

In the prospective validation cohort (tissues), 25 LUAD patients were enrolled from Peking University People's Hospital, and tumor tissues in situ and adjacent normal lung tissues were obtained during surgery. The pathological diagnosis of tumor samples was completed by two pathologists. Molecular pathology was performed at the molecular pathology testing center of Peking University People's Hospital. The clinicopathological information of subjects in this study is summarized in Appendix Table S3.

In each cohort, Chi-square test was adopted for the analysis of baseline characteristics of the study population, and Mann–Whitney $U$ test was used to determine the differences between groups.

## Untargeted lipid profiling

Lipids were extracted from plasmas as previously described (Zhou et al, 2017). Briefly, 100 μL of sample was transferred to an EP tube, and added with 480 μL of extract solution (MTBE: methanol1 = 5: 1 with internal standard). After 30 s vortex, the samples were sonicated for 10 min in ice-water bath. Then the samples were incubated at −40 °C for 1 h and centrifuged at 3000 rpm for 15 min at 4 °C. In total, 350 μL of supernatant was transferred to a fresh tube and dried in a vacuum concentrator at 37 °C. Then, the dried samples were reconstituted in 100 μL of 50% methanol in dichloromethane by sonication for 10 min in ice-water bath. The constitution was then centrifuged at 13,000 rpm for 15 min at 4 °C, and 80 μL of supernatant was transferred to a fresh glass vial for LC/MS analysis. The quality control (QC) sample was prepared by mixing an equal aliquot of the supernatants from all of the samples.

LC–MS/MS analyses were performed using an UHPLC system (1290, Agilent Technologies), equipped with a Kinetex C18 column (2.1 × 100 mm, 1.7 μm, Phenomen). The mobile phase A consisted of 40% water, 60% acetonitrile, and 10 mmol/L ammonium formate. The mobile phase B consisted of 10% acetonitrile and 90% isopropanol, which was added with 50 mL 10 mmol/L ammonium formate for every 1000 mL mixed solvent. The analysis was carried with elution gradient as follows: 0–12.0 min, 40–100% B; 12.0–13.5 min, 100% B; 13.5–13.7 min, 100–40% B; 13.7–18.0 min, 40% B. The column temperature was 55 °C. The auto-sampler temperature was 4 °C, and the injection volume was 2 μL (pos) or 4 μL (neg), respectively. The QE mass spectrometer was used for its ability to acquire MS/MS spectra on data-dependent acquisition (DDA) mode in the control of the acquisition software (Xcalibur 4.0.27, Thermo). In this mode, the acquisition software continuously evaluates the full-scan MS spectrum. The ESI source conditions were set as follows: sheath

gas flow rate as 30 Arb, Aux gas flow rate as 10 Arb, capillary temperature 320 °C (positive), 300 °C (negative), full MS resolution as 70,000, MS/MS resolution as 17,500, collision energy as 15/30/45 in NCE mode, spray voltage as 5 kV (positive) or −4.5 kV (negative), respectively.

The raw data files were converted to files in mzXML format using the "msconvert" program from ProteoWizard. Peak detection was first applied to the MS1 data. The CentWave algorithm in XCMS was used for peak detection with the MS/MS spectrum, lipid identification was achieved through a spectral match using LipidBlast library.

## Targeted lipid profiling

For plasma samples, lipids were extracted as follows. In total, 10 μL of the sample was mixed with 190 μL water, and then 480 μL extract solution (MTBE: methanol = 5:1) containing internal standard was added. After 60 s vortex, the samples were sonicated for 10 min in ice-water bath. Then the samples were centrifuged at 3000 rpm for 15 min at 4 °C. 250 μL of supernatant was transferred to a fresh tube. The rest of the sample was added with 250 μL of MTBE, followed with vortex, sonication and centrifugation, and another 250 μL of supernatant was taken out. This step was repeated once. And the supernatants were combined and dried in a vacuum concentrator at 37 °C. Then, the dried samples were reconstituted in 100 μL of resuspension buffer (DCM: MeOH: $H_2O$ = 60:30:4.5), the samples were vortexed for 30 s and sonicated for 10 min in ice-water bath. The constitution was then centrifuged at 12,000 rpm for 15 min at 4 °C, and 30 μL of supernatant was transferred to a fresh glass vial for LC/MS analysis. The quality control (QC) sample was prepared by mixing an equal aliquot of the supernatants from all of the samples.

For tissue samples, lipids were first extracted. Briefly, 10 mg of the sample was weighted on the dry ice to an EP tube, After the addition of 400 μL water, the samples were vortexed for 60 s, homogenized at 45 Hz for 4 min, and sonicated for 5 min in ice-water bath. The homogenate and sonicate circle was repeated for three times. The 10 μL homogenate was mixed with 190 μL water, and then 480 μL extract solution containing internal standard was added. After vortexing for 60 s, the samples were sonicated for 10 min in ice-water bath. Then the samples were centrifuged at 3000 rpm for 15 min at 4 °C. Overall, 250 μL of supernatant was transferred to a fresh tube. The rest of the sample was added with 250 μL of MTBE, followed with vortex, sonication and centrifugation, and another 250 μL of supernatant was taken out. This step was repeated once. And the supernatants were combined and dried in a vacuum concentrator at 37 °C. Then, the dried samples were reconstituted in 200 μL of resuspension buffer (DCM:MeOH:$H_2O$ = 60:30:4.5), the samples were vortexed for 30 s and sonicated for 10 min in ice-water bath. The constitution was then centrifuged at 12,000 rpm for 15 min at 4 °C, and 40 μL of supernatant was transferred to a fresh glass vial for LC/MS analysis. The quality control (QC) sample was prepared by mixing an equal aliquot of the supernatants from all of the samples.

For cell lines, all samples from cell lines were mixed with 200 μL water, the samples were vortexed for 60 s, freeze and thaw three times with liquid nitrogen, and sonicated for 2 min in ice-water bath. The 200 μL homogenate was mixed with 480 μL extract solution (MTBE: MeOH = 5: 1) containing internal standard was

added. After 60 s vortex, the samples were sonicated for 10 min in ice-water bath. Then the samples were centrifuged at 3000 rpm for 15 min at 4 °C. 250 μL of supernatant was transferred to a fresh tube. The rest of the sample was added with 250 μL of MTBE, followed with vortex, sonication and centrifugation, and another 250 μL of supernatant was taken out. This step was repeated once. And the supernatants were combined and dried in a vacuum concentrator at 37 °C. Then, the dried samples were reconstituted in 80 μL of resuspension buffer (DCM: MeOH: $H_2O$ = 60: 30: 4.5), the samples were vortexed for 30 s and sonicated for 10 min in ice-water bath. The constitution was then centrifuged at 12,000 rpm for 15 min at 4 °C, and 35 μL of supernatant was transferred to a fresh glass vial for LC/MS analysis. The quality control (QC) sample was prepared by mixing an equal aliquot of the supernatants from all of the samples.

LC–MS/MS analyses were then performed. The UHPLC separation was carried out using a SCIEX ExionLC series UHPLC System. The mobile phase A consisted of 40% water, 60% acetonitrile, and 10 mmol/L ammonium formate. The mobile phase B consisted of 10% acetonitrile and 90% isopropanol, and 10 mmol/L ammonium formate. The column temperature was 40 °C. The auto-sampler temperature was 6 °C, and the injection volume was 2 μL. AB Sciex QTrap 6500+ mass spectrometer was applied for assay development. Typical ion source parameters were: IonSpray Voltage: +5500/ −4500 V, Curtain Gas: 40 psi, Temperature: 350 °C, Ion Source Gas 1:50 psi, Ion Source Gas 2: 50 psi, DP: ±80 V.

Skyline 20.1 Software was employed for the quantification of the target compounds. The absolute content of individual lipids corresponding to the IS was calculated on the basis of peaks area, and actual concentration of the identical lipid class internal standard (IS), and then absolute content was obtained from diverse internal standard (IS) averaged of the identical lipid class.

## Identification of signature lipids from untargeted lipid profiling

SIMCA-P 14.1 (Umetrics, Umca, Sweden) was employed for the orthogonal partial least squares discriminant analysis (OPLS-DA) model, which was used to understand global lipid changes between LUAD patients and HC samples, and corresponding variable importance in the projection (VIP values) was calculated as well. A permutation test was used based on the OPLS-DA model to estimate the robustness and the predictive ability of our model. Mann–Whitney $U$ test and multiple comparison correction were applied upon the normalized untargeted lipid data to identify differential lipids between LUAD and HCs. Differential lipids were determined with VIP>1, FDR adjust $P$ value <0.05 and fold change threshold 1.5. In total, 2000 iterations of random sampling were then applied on the untargeted lipid data of differential lipids, in which 90 samples were randomly divided into a training set (70% of all samples) and a test set (30% of all samples). For each iteration of random sampling, Random Forest and XGBoost were used to deduce signature lipid candidates which were able to classify LUADs from HCs, respectively. For the signature lipid candidates derived by Random Forest, those with the average importance more than 0.7 were selected as the potential signature lipids. For the signature lipid candidates derived by XGBoost, those with the number of occurrences more than 1200 out of 2000 iterations were selected as the potential signature lipids. Lasso was also used upon

all samples to infer potential signature lipids from differential lipids. The untargeted lipid data of differential lipids in positive and negative ion modes were analyzed separately. All potential signature lipids consistently inferred by all the three methods including Random Forest, XGBoost, and Lasso were determined as the final signature lipids. Hierarchical clustering and ROC analysis were performed to further evaluate the accuracy and specificity of the signature lipids in classifying LUADs from HCs.

## LSRscore construction and validation

Among the signature lipids identified from untargeted lipid profilings, only those having internal standards available were remained, resulting in four signature lipids. ROC analysis was performed upon all samples in untargeted lipid profiling to confirm the reliability of predicting LUADs for each of the four signature lipids, respectively. A lipid signature-based scoring model (LSRscore) was developed to score each sample by the sum of the concentration of two upregulated signature lipids divided by the sum of the concentration of two downregulated ones.

Targeted lipidomics assay was performed upon the prospective validation cohort of plamas from LUADs for the four signature lipids. LSRscore was then calculated. ROC analysis and Multivariate binary logistic analysis was performed to validate the predictive power of LSRscore for LUADs.

## Evaluation of signature lipids in LUAD cancer tissues in situ

Targeted lipidomics data of PE(18:0/18:1) and PC(18:2/18:2) were generated as above, and Mann–Whitney $U$ test and Paired $T$ test were applied to compare the concentration difference between LUAD cancer tissue and adjacent normal lung tissues, respectively. $P$ value ≤ 0.05 was determined as significance. ROC analysis was applied upon targeted lipidomics data of cancer and adjacent normal lung tissues to evaluate the predictive power of PE(18:0/18:1) and PC(18:2/18:2) for LUADs.

## Single-cell RNA-seq data processing

scRNA-seq data of healthy lung tissues from eight lung transplant donors were downloaded from GEO database (GSE122960) (Data ref: Reyfman et al, 2019). scRNA-seq data of early-stage LUAD were collected from GEO database (GSE131907) (Data ref: Kim et al, 2020) and public domain(Data ref: Bischoff et al, 2021) with samples from stage III/IV excluded from the downstream analysis (9 tumor and 9 adjacent normal tissues in nLung or tLung groups from stage III/IV samples excluded for GSE122960, and 8 tumor and 8 adjacent normal tissues from stage III/IV samples excluded for scRNA-seq data from Bischoff's article). Genes expressed in more than 10 samples and cells with >200 genes detected were remained, followed by removal of low-quality cells (<500 UMIs; >10,000 or <600 genes; >10% UMIs derived from the mitochondrial genome). The gene expression matrices were normalized by the *NormalizeData* function in Seurat package of R. The most variable 2000 genes were determined by the *FindVariableGenes* function with default parameters. PCA was performed using *RunPCA* with these 2000 genes in Seurat. *RunHarmony* Harmony function was then applied to remove the effects of confounding factors across different samples. Dimensionality reduction was performed using PCA and UMAP plots were generated by *RunUMAP* Seurat function with the first 30 PCs as input. *Clustree* package was used to find the appropriate resolution. *FindNeighbors* and *FindClusters* function in Seurat with *FindClusters* parameter "resolution = 0.6" were used to identify cell subpopulation, obtaining 29 clusters.

Based on the DEGs (differentially expressed genes) for each cluster, the first round of clustering was performed for the identification of major cell types including T & NK cells, B cells, myeloid cells and endothelial cells, fibroblast cells, and epithelial cells. An additional round of clustering was performed within each major cell type. The resulting clusters with more than 15% of cells with doublet (defined by DoubletFinder v2.0.3) (McGinnis et al, 2019) were flagged, and the cells would be identified as doublet if they expressed more than one set of canonical markers. Besides, cells with Epithelial markers and $PTPRC^+$ were also identified as doublet. Finally, 44,480 nonimmune cells were retained for further analysis.

## Tumor cell copy number inferring

inferCNV was used to predict the copy number alteration with the default parameters, and immune cells in LUAD samples were considered as putative nonmalignant cells and their CNV estimates were used to define baseline. CNVcor was calculated, which was defined as the Pearson correlation coefficient between each cell's CNVscore and the average CNVscore of the top 10% of cells. Data from the epithelial and stromal cells in lung cancer samples were inputted as malignant cells, which were classified with CNVscore more than mean+sd of healthy group CNVscore and CNVcor >0.2.

## Cell-type annotation

*FindAllMarkers* function in Seurat was applied to find differentially expressed genes for each cell subpopulation. Cell subpopulations were then classified and annotated based on the expression of canonical markers of different cell types. B cells, myeloid cells, endothelial cells, fibroblast cells, T & NK cells, epithelial cells, and cycling cells were identified. SingleR and SciBet (Li et al, 2020) were also used to for cell-type annotation.

## GO enrichment analysis for scRNA-seq data

FindMarkers function was used to identify differentially expressed genes (DEGs) in scRNA-seq data, and DEGs were filtered using a minimum avg_logFC of 0.25 and FDR value of 0.05. GO enrichment analysis for the functions of the DEGs was conducted using R package of clusterProfiler (Yu et al, 2012). Gene sets with FDR ≤ 0.05 were determined as significantly enriched. Cytoscape (Shannon et al, 2003) were applied to visualize the enriched gene sets.

## Cell culture

Human normal lung epithelial cells (BEAS-2B) were purchased from Servicebio (catalog number: STCC10202G), and human lung adenocarcinoma cells (A549) were purchased from Shanghai Zhong Qiao Xin Zhou Biotechnology Co., Ltd. (catalog number: ZQ0003). The cell lines were recently authenticated by STR profiling. BEAS-2B/A549 cells were cultured in DMEM complete medium containing DMEM basic medium (Servicebio), 10% fetal

bovine serum (Gibco), and 1% penicillin–streptomycin (Gibco) at 37 °C with 5% $CO_2$.

The cell medium was removed and cells were washed quickly with pre-cooled PBS solution with twice repeaed, then remove the PBS solution. Trypsin digestion was added until some of the cells were spherical. Add the medium and shake gently to terminate the digestion of infiltrated cells and aspirate the remaining medium. After the addition of PBS, suspend the cells to digest the adherent cells. One volume of cells was taken by rapid counting at 4 °C, centrifuged at $2500 \times g$ for 5 min, supernatant removed, and quickly stored in liquid nitrogen at −80 °C for targeted lipidomics measurements.

After the culture medium was removed, the cells were washed with PBS, one hole of a six-well plate was added with 1 ml Trizol, blown for full digestion, transferred to 1.5 ml/2 ml EP tube of RNasefree, stored at −80 °C, and transcriptomic measurements were performed.

### Cell culture of LUAD, normal, and liver carcinoma cell lines

Human skin fibroblasts (HSF) were obtained from Prof. Youchun Xu group at Tsinghua University and human liver carcinoma cells (HepG2) as a control were obtained from Prof. Xingjie Liang group at the National Center for Nanoscience and Technology, China. LUAD cells including A549 cells and NCI-H1975 (Servicebio, catalog number: STCC10204P). A549/BEAS-2B/HepG2 cells were cultured in DMEM complete medium containing DMEM basic medium (Servicebio), adding 10% fetal bovine serum (Gibco) and 1% penicillin–streptomycin (Gibco) at 37 °C with 5% $CO_2$, while HSF/NCI-H1975 cells were cultured in RPMI-1640 complete medium containing, adding 10% fetal bovine serum (Gibco), and 1% penicillin–streptomycin (Gibco) at 37 °C with 5% $CO_2$. Cells were passaged when they grew to about 80%. The cell medium were washed with PBS and trypsin digestion was added for 2 min, then, taken 15 ml into a centrifuge tube at 1000 rpm for 3 min. The supernatant was discarded, resuspended with the corresponding medium and added to new culture dish. Preparation for targeted lipidomics measurements according to the method above.

### Cell lines treated with dose-dependent of PE(18:0/18:1)

PE(18:0/18:1) was purchased from Sigma-Aldrich (product num. 850758P).1.66 mg of PE was taken into a 5 ml EP tube, 500 µl of trichloromethane was added, after which the trichloromethane was evaporated with nitrogen, at which time the PE(18:0/18:1) would leave a film in the EP tube, followed by the addition of 1.1 ml of 0.1% ethanol solution, and ultrasonication was performed for 1 min to form a suspension, at which time the concentration of PE(18:0/18:1) was 2 mmol/L (PE(18:0/18:1) mother solution). BEAS-2B cells were laid out on six-well plates and cultured for 1 day, then dosed so that the final concentration of PE(18:0/18:1) was 100 µmol/L, 10 µmol/L, 1 µmol/L, and 0 µmol/L, respectively After 26 h of culture, the transcriptomics preparation was carried out: the culture medium was removed, the cells were washed with PBS, 1 ml of Trizol was added to one well of the six-well plate, blown and fully digested, transferred to RNase-free 1.5 ml EP tubes, and stored at −80 °C and transported on dry ice for transcriptome measurements.

### The NES value for each metabolic pathway in RNA-Seq

The KEGG ID of PE(18:0/18:1) was obtained from the LIPID MAPS Structure Database and MetaboAnalyst, and the associated pathways from the KEGG PATHWAY Database to obtain the corresponding pathways. KEGGREST package in R provides a client interface to the Kyoto Encyclopedia of Genes and Genomes (KEGG) REST server, and the gene list in each metablic pathway were obtained.

To evaluate the changes enrichment of each metabolic pathway related in TCGA-LUAD and CPTAC-LUAD data, we used Normalized Enrichment Score (NES) (Zhang et al, 2019), which is an estimate of the probability that the expression of a gene in the gene set is greater than the expression of a gene outside this set.

$$\text{NES} = 1 - \frac{U}{mn}$$

$$U = mn + \frac{m(m+1)}{2} - T$$

where m is the number of genes in a gene set, $n$ is the number of those outside the gene set, and $T$ is the sum of the ranks of the genes in the gene set.

### RNA-seq data collection

RNA-seq raw data of 557 and 229 LUADs were collected from TCGA-LUAD and CPTAC-LUAD, respectively. Only the primary tumor samples (01 A) were retained, resulting in 487 samples in TCGA-LUAD. RNA-seq read count data of 120 normal lung tissues were collected from public databases, including 26 samples from GSE134692, 86 samples from GSE165192, and 8 samples from GSE120795.

### RNA-seq data analysis

RNA-seq was performed of 6 subjects (3 subjects with LUAD A549 cell line and 3 subjects with Healthy lung epithelial BEAS-2B celline) and 12 subjects (3 subjects of LUAD A549, 3 subjects of LUAD NCI-H1975, 3 subjects of BEAS-2B, 3 subjects of HSF, and 3 subjects of HepG2). The 7 G raw FASTQ data per sample were generated by the Illumina HiSeq 2000 platform. FASTQ data were aligned to the human reference genome (hg19) using the STAR (Dobin et al, 2013) at default settings, and Htseq-count (Anders et al, 2015) was used to compute read counts for each gene.

### Pathway enrichment analysis

Pathway enrichment analysis was conducted upon DEGs using R package of clusterProfiler (Yu et al, 2012). Pathways included PE(18:0/18:1) were extracted from KEGG database. Pathways with FDR ≤ 0.05 were determined as significantly enriched.

The enrichment status of each pathway incorporated PE(18:0/18:1) was evaluated by GSVA for each patient in CPTAC-LUAD,TCGA-LUAD and cell lines, respectively. For each pathway, median value of normalized enrichment score (NES) was used to divide patients into two groups. Kaplan–Meier analysis was performed to evaluate the survival differences between two groups. Multivariate Cox regression analyses were performed to evaluate whether the pathway enrichment score was associated with overall survival independent of other clinical factors.

### The paper explained

**Problem**

Lung adenocarcinoma (LUAD) is the most common histological type of lung cancer. Although significant progress has been made in its treatment, LUAD continues to have a high mortality rate and lacks reliable biomarkers for noninvasive early cancer diagnosis. To improve the overall survival of patients with LUAD, the establishment of reliable biomarkers for early-stage LUAD diagnosis is urgently needed.

**Results**

Our study reveals significant abnormalities in lipid metabolism within lung cancer cells and identifies four distinct lipid biomarkers. Subsequently, we introduce a plasma lipid signature-based scoring model (LSRscore) for early cancer detection, achieving an AUC of 0.972. Validation confirms strong predictive capability of LSRscore for LUAD, with an AUC of 0.92. In addition, we observe consistent alterations of PE(18:0/18:1) levels in both cancer tissue and plasma, demonstrating robust predictive value for early-stage LUAD. Increased levels of PE(18:0/18:1) are associated with dysregulated glycerophospholipid metabolism, showing reliable prognostic power across two LUAD cohorts.

**Impact**

LSRscore and PE(18:0/18:1) exhibit reliable predictive accuracy for early-stage LUAD. Consistently increased levels PE(18:0/18:1) disrupting glycerophospholipid metabolism offer significant prognostic capacity. These signature lipid biomarkers and LSRscore hold promise for early-stage diagnosis and prognosis of LUAD.

## Statistics

Comparisons between two groups were performed using an unpaired two-tailed Mann–Whitney $U$ test, paired $t$ test or $t$ test with Welch correction (two-tailed, unequal variance) when appropriate. Survival curves were compared using a log-rank test (Mantel–Cox). Categorical variables were compared using one-sided or two-sided Fisher exact test. Multivariate survival analysis was performed using Cox regression model.

## For more information

Data websites: Lipidomics data: https://ngdc.cncb.ac.cn/omix. TCGA-LUAD: https://portal.gdc.cancer.gov/repository. CPTAC-LUAD: https://portal.gdc.cancer.gov/. Lipid website: KEGG PATHWAY Database: https://www.genome.jp/kegg/pathway.html. LIPID MAPS: https://lipidmaps.org/. MetaboAnalyst: https://www.metaboanalyst.ca/.

## Data availability

The datasets produced in this study are available in the following databases: Raw lipidomics data: OMIX, China National Center for Bioinformation/Beijing Institute of Genomics, Chinese Academy of Sciences (https://ngdc.cncb.ac.cn/omix) under accession no. OMIX002792. Custom codes and data: Biostudies (https://www.ebi.ac.uk/biostudies/) with the accession number of S-BSST1198

(https://www.ebi.ac.uk/biostudies/studies/S-BSST1198?key=277efffe-1d1a-41d5-a988-4b098e28d670).

## Peer review information

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

## Acknowledgements

This work was supported by grants from Beijing Hospital Authority Clinical Medicine Development special funding support, code ZLRK202333 (JZ), the Youth Thousand Scholar Program of China (JZ), Fundamental Research Funds for the General Universities (JZ), the National Natural Science Foundation of China (grant no. 82202336, JGC), Guangdong Basic and Applied Basic Research Foundation (grant no. 2021A1515110619, JGC), Clinical Medicine Plus X-Young Scholars Project of Peking University (PKU2023LCXQ001, XW), and the Beijing Natural Science Foundation (L222097 to JZ). The funders had no

role in study design, data collection and analysis, decision to publish, or preparation of the manuscript. We thank BIOTREE for assistance with lipidomics assay and Tsingke Biotechnology for assistance with RNA sequencing. We thank W Shi and J Yu for the critical discussion. The authors also thank Taicang Qingli Biotechnology Co., Ltd, and Ling Wei Pharmaceutical Inc for providing valuable comments and discussion about bioinformatics analysis and experiments related to this work.

## Author contributions

**Ting Sun**: Data curation; Formal analysis; Investigation. **Junge Chen**: Conceptualization; Formal analysis; Funding acquisition; Writing—original draft; Writing—review and editing. **Fan Yang**: Resources; Investigation. **Gang Zhang**: Formal analysis; Investigation. **Jiahao Chen**: Validation; Writing—review and editing. **Xun Wang**: Resources; Funding acquisition; Investigation. **Jing Zhang**: Conceptualization; Supervision; Funding acquisition; Investigation; Writing—original draft; Project administration; Writing—review and editing.

## Disclosure and competing interests statement

The authors declare no competing interests.

