## [Peer Review File · EMBO Molecular Medicine]

Lipidomics reveals new lipid-based lung adenocarcinoma early diagnosis model

Ting Sun, Junge Chen, Fan Yang, Gang Zhang, Jiahao Chen, Xun Wang, and Jing Zhang

Corresponding authors: Jing Zhang (jz2716@buaa.edu.cn) , Xun Wang (wangxun04275@pkuph.edu.cn)

Review Timeline:

Submission Date:	18th May 23
Editorial Decision:	21st Jun 23
Revision Received:	21st Sep 23
Editorial Decision:	26th Oct 23
Revision Received:	24th Jan 24
Editorial Decision:	13th Feb 24
Revision Received:	21st Feb 24
Accepted:	27th Feb 24

Editor: Zeljko Durdevic

Transaction Report:

21st Jun 2023

Dear Dr. Zhang,

Thank you for the submission of your manuscript to EMBO Molecular Medicine. We have now received feedback from the three reviewers who agreed to evaluate your manuscript. All three referees recognize potential interest of the study but also raise important and partially overlapping criticism that should be addressed in a major revision. Particular attention should be given to providing further mechanistic insights of the connection of PE(18:0/18:1) with dysregulated glycerophospholipid metabolism in the context of LUAD as suggested by the referee #2. Further, detailed description of statistical tests used, randomization, blinding, replication etc. should be provided in Materials and Methods section. I would also recommend running the article by a native English speaker. If you would like to discuss further the points raised by the referees, I am available to do so via email or video. Let me know if you are interested in this option.

We would welcome the submission of a revised version within three months for further consideration. Please let us know if you require longer to complete the revision.

I look forward to receiving your revised manuscript.

Yours sincerely,

Zeljko Durdevic

We require:

- 1) A .docx formatted version of the manuscript text (including legends for main figures, EV figures and tables). Please make sure that the changes are highlighted to be clearly visible.
- 2) Individual production quality figure files as .eps, .tif, .jpg (one file per figure). For guidance, download the 'Figure Guide PDF': (<https://www.embopress.org/page/journal/17574684/authorguide#figureformat>).
- 3) A .docx formatted letter INCLUDING the reviewers' reports and your detailed point-by-point responses to their comments. As part of the EMBO Press transparent editorial process, the point-by-point response is part of the Review Process File (RPF), which will be published alongside your paper.
- 4) A complete author checklist, which you can download from our author guidelines (<https://www.embopress.org/page/journal/17574684/authorguide#submissionofrevisions>). Please insert information in the checklist that is also reflected in the manuscript. The completed author checklist will also be part of the RPF.
- 5) Please note that all corresponding authors are required to supply an ORCID ID for their name upon submission of a revised

manuscript.

6) It is mandatory to include a 'Data Availability' section after the Materials and Methods. Before submitting your revision, primary datasets produced in this study need to be deposited in an appropriate public database, and the accession numbers and database listed under 'Data Availability'. Please remember to provide a reviewer password if the datasets are not yet public (see <https://www.embopress.org/page/journal/17574684/authorguide#dataavailability>).

13) Author contributions: You will be asked to provide CRediT (Contributor Role Taxonomy) terms in the submission system. These replace a narrative author contribution section in the manuscript.

14) A Conflict of Interest statement should be provided in the main text.

Please note: When submitting your revision you will be prompted to enter your funding and payment information. This will allow Wiley to send you a quote for the article processing charge (APC) in case of acceptance. This quote takes into account any reduction or fee waivers that you may be eligible for. Authors do not need to pay any fees before their manuscript is accepted and transferred to the publisher.

EMBO Press participates in many Publish and Read agreements that allow authors to publish Open Access with reduced/no publication charges. Check your eligibility: <https://authorservices.wiley.com/author-resources/Journal-Authors/open-access/affiliation-policies-payments/index.html>

***** Reviewer's comments *****

Referee #1 (Remarks for Author):

In this article Wang and colleagues have performed untargeted lipidomic profiling of lipids in plasma obtained from lung adenocarcinoma (LUAD) patients and healthy controls. This approach identified a 4-lipid signature with potential diagnostic power that was subsequently confirmed by targeted lipidomic assay in a validation cohort. This concerted action led to the development of a lipid signature-based ratio scoring model (LSRscore) that outperformed 3 of the 4 lipid individual signatures. Furthermore, targeted lipidomics in surgically resected tissues validated increased levels of PE(18:0/18:1) in tumours when compared with adjacent tissues. ,

The lipidomic studies are supported by transcriptomic analyses (both bulk RNA seq and single cell RNA seq) not performed by the authors but obtained from available public repositories. Bioinformatic analyses confirm that altered glycerophospholipid metabolism is altered in LUAD.

I feel that a number of additional questions should be addressed.

1. This study is technically and conceptually similar to a recent report cited by the authors (Wang et al doi: 10.1126/scitranslmed.abk2756) that performed a technically similar approach (LC-MS-based targeted lipidomics assay using MRM model). This manuscript described a 9 lipid diagnostic signature of which only 4 -including the most precise PE(18:0/18:1)- were identified as differentially present by the present study. The authors should discuss the potential causes underlying this discrepancy and the potential implications for the general implementation of their own signature.
2. It is unclear for the non-expert to what extent a targeted lipidomic assay would outperform other non-invasive diagnostic approaches such as ctDNA. The authors should comment on this.
3. Similarly, it is not entirely clear how this targeted lipidomic assay would be LUAD specific and not result in cross-diagnostic of other tumour types. As an example, a recent serum lipidomic profiling for pancreatic ductal adenocarcinoma (doi: 10.1038/s41467-021-27765-9) reported a lipid signature that also includes PE(18:0/18:1). This fact should be discussed (and the manuscript cited).
4. There are increasing evidences suggesting that LUAD displays substantial heterogeneity in energy metabolism within and between tumours (doi: 10.1016/j.cell.2015.12.034). Does the current work suggest that lipid metabolism is more homogeneous? The authors should at least address this question between patients.
5. Finally, I am unsure as to whether using publicly available data merits the inclusion of scRNA sequencing as part of the manuscript title.

Referee #2 (Comments on Novelty/Model System for Author):

As stated in my report I have some concerns about the equity and characterization of the data in the different groups that were compared in this study. The authors must provide additional information, as requested. However, I think this the proposed plasma lipid signature-based scoring model developed with human samples is interesting, and the poor survival rates and outcomes in lung cancer worldwide justify the importance of designing and validating new early detection tools.

Therefore, I would reconsider the paper after major revisions. Considering the high impact and quality of EMBO Mol Med I find crucial to address three crucial points: (i) provision of additional information regarding the transcriptomic and lipidomic analyses (ii) a better assessment/reanalysis of the LSRscore in clinical cohorts (e.g. considering treatment status of the patients) (iii) to provide further mechanistic insights of the connection of PE(18:0/18:1) with dysregulated or rewiring glycerophospholipid metabolism.

Referee #2 (Remarks for Author):

In this manuscript entitled "Single cell RNA-seq and lipidomics reveals new lipid-based lung adenocarcinoma early diagnosis model", Sun et al. propose a lipid signature biomarker for early diagnosis of lung adenocarcinoma. The authors show evidence by scRNAseq of aberrant lipid metabolism associated with lung cancer cells and identify 4 signature lipid biomarkers by profiling lipidomics in patient samples. Based on this, they develop a plasma lipid signature-based scoring model that they call LSRscore. The authors propose that LSRscore can predict lung adenocarcinoma in a discovery cohort and an independent validation cohort with AUC >0.9. In addition, the authors show that PE(18:0/18:1) lipid biomarker is altered in plasma and tumour tissue thereby being suitable for prediction purposes, and suggest that it is associated with dysregulated glycerophospholipid metabolism.

Overall, the manuscript is interesting, and the developed methodology warrants further investigations due to the unmet clinical need of novel and more refined tools for lung cancer early detection. Such a tool would be of tremendous importance considering the current poor survival rates of lung cancer patients and the significant curable rates when the disease is detected at early stages. However, the existing data do not fully support the claims of the authors in this manuscript and they must clarify some of aspects of their experimental designs and the used databases to validate the efficacy and utility of their lipid signature-based model, see my main points below. The validation of the LSRscore with the independent cohort is questionable. Also, additional mechanistic insights about the interconnection of PE(18:0/18:1) with dysregulated glycerophospholipid metabolism must be provided. Overall, I consider there are major concerns as outlined below that should be addressed to meet with the high quality standards of EMBO Mol Med.

Major concerns:

1. A main point in this study is that both scRNAseq data and RNAseq data have been downloaded from GEO (GSE122960, GSE131907) and TCGA-LUAD/CPTAC-LUAD, but it is unclear the methodology, custom code and raw data used in this study. This information should be clearly provided and deposited in an appropriate repository. Also, it must be clear by referencing in the text (in the Results section) that these are not original data from the authors but collected from databases/publications.
2. Following my previous comment, the data provided in Figure 1 seem uncomplete and difficult to confirm/corroborate. Marking scores (scales) are missing in the graphs and also the enrichment score of the GSEA (e.g. Fig. 1F).
3. The data presented in the Discovery Cohort make the conclusions speculative and lack of robustness. In particular because of the significant differences between the healthy subjects and LUAD patients in gender and age (i.e. average of 29 years old in the healthy cohort and 59 years old in the LUAD patients). Even for the validation cohort the differences in age (average of 51 years old healthy individuals versus 62 years old LUAD patients) make the conclusions of the authors should be considered carefully. Metabolic and lipidomic profiles can be significantly altered/distinct with age.
4. Figure 3C-H. The data of the LSRscore in predicting LUAD with the Independent cohort seem exciting. However, the clinical characteristics of these patients is missing. This information should be provided in the Supplementary Material. For example, are all of these patients treatment naïve? Lipidomic profiles can be altered by treatment conditions. Also, I wonder whether the authors could clarify why the levels of the four lipids in plasma from LUADs and HCs are measured in logarithmic₂ scale in Fig 3E-H but not in Fig 2F-I. Checking carefully the information provided in this Figure and considering the scale of the Y axis the differences seem really minimal and hence the validation is questionable.
5. Figure 5. Coming back to my Point 1, the results shown in Figure 5 need of additional information to be corroborated. All the information regarding clinical characteristics is missing in the Supplementary Information. Again, the information in the Cancer Genome Atlas is frequently incomplete in terms of these characteristics, for example whether the patients were or not treated and when. This knowledge is very important as per the potential effects of cancer therapies in the glycerophospholipid

metabolism.

6. The authors claim that the overexpression of PE(18:0/18:1) is associated with the dysregulated glycerophospholipid metabolism in the context of LUAD. The results and this hypothesis are stimulating but the authors must provide some mechanistic insights to support this claim to exclude other factors or pathological conditions, such as ageing, inflammation, etc. I find essential to provide a clear evidence of the connection of PE(18:0/18:1) with dysregulated or rewiring glycerophospholipid metabolism for example by using LUAD cell lines. In summary, it is not clear whether upregulated levels of PE(18:0/18:1) are directly or causally connected with LUAD or just indirectly correlated.

Other points/suggestions:

1. English grammar must be revised throughout the text.
2. Supplementary Figure 2. Why the training set was done with 70% of the samples and only a 30% of the samples was used for the test set?
3. Fig. 3E-H. LUADs and HCs are inverted (left and right) in the graph from Figs. 2 and 3, this is confusing. This also happens in Fig. 4 A-C versus Fig. 4B-D.

Referee #3 (Comments on Novelty/Model System for Author):

The manuscript and data may need evaluation from statistic experts.

Referee #3 (Remarks for Author):

The manuscript by Sun et al aimed to develop lipid-based biomarkers for non-invasive diagnosis of early lung adenocarcinoma (LUAD). They found that aberrant lipid metabolisms significantly enriched in lung cancer cells by analyzing single cell RNA-sequencing data from public domain and then identify 4 signature lipid biomarkers by profiling lipidomics of LUADs. They then develop a plasma lipid signature-based scoring model (LSRscore) and that LSRscore has a powerful prediction of LUAD in discovery cohort of 90 subjects (with AUC reaching 0.972) and independent validation cohort of 60 participants (with AUC of 0.924). They demonstrated that a lipid signature biomarker PE(18:0/18:1), is consistently altered in both cancer tissues in situ and plasmas of 25 LUADs, showing strong predictive power for early-stage LUAD. In the transcriptome analysis of 421 LUAD cancer tissues and 120 normal lung tissues show that PE(18:0/18:1) overexpression is associated with dysregulated glycerophospholipid metabolism, which consistently exhibit great prognostic power in two large LUAD cohorts. They suggested that LSRscore and PE(18:0/18:1) may be useful for early-stage diagnosis and survival prediction of LUADs.

The manuscript may be intriguing and the lipid-based biomarkers may be useful if the results can be validated in different institute with larger non-lung cancer cohort.

They are several comments that need to be addressed.

1. The number of normal controls may be too limited, which did not simulated in real world condition; and was not stratified smoking status and other lung conditions, such as COPD, pneumonia and others. it is not known whether the lipid-based biomarkers still workable in inflammatory lung diseases and whether false positive may occur in condition of acute lung injury-repair.
2. Although the authors claimed that the LSRscore may be helpful in clinical applications to improve clinical detection and screening for LUAD without invasive diagnostic procedures or unnecessary exposure to radiation as required by LDCT. If they can demonstrate that combination of lipid-based assay and LDCT can further improve the diagnostic accuracy, the study will be even more powerful.
3. In line 6 of introduction, the citation of 5-year survival should be better include both overall and stage 1. The citation of stage 1 survival needs to be updated to the most recent one. The 5-year survival of 60% in stage 1 disease seems not meet the current standard of care.
4. In the result first paragraph, "To discover reliable biomarkers for early-stage lung adenocarcinoma (LUAD) diagnosis, we collected single cell RNA sequencing (scRNA-seq) data of tumor and adjacent normal tissues from 17 LUADs and eight healthy lung samples from healthy donors in the public domain." The source of public domain should be cited. The demographic information including age, gender, smoking status, stage and driver gene status should be described. How these 17 LUAD were selected?
5. In supplementary table 1 which summarized the demographic information of study population. The readers will be interested

to know in more detail the smoking status and underlying diseases in normal subjects. It is also important to know if the lipid profile altered in different driver gene mutations, such as EGFR and Kras, especially EGFR which is nearly 50% in LUAD in China, it may be important to know whether the lipid-based biomarkers can be applied independent of all different drivers.

***** Reviewer's comments *****

Referee #1 (Remarks for Author):

In this article Wang and colleagues have performed untargeted lipidomic profiling of lipids in plasma obtained from lung adenocarcinoma (LUAD) patients and healthy controls. This approach identified a 4-lipid signature with potential diagnostic power that was subsequently confirmed by targeted lipidomic assay in a validation cohort. This concerted action led to the development of a lipid signature-based ratio scoring model (LSRscore) that outperformed 3 of the 4 lipid individual signatures. Furthermore, targeted lipidomics in surgically resected tissues validated increased levels of PE(18:0/18:1) in tumours when compared with adjacent tissues.

The lipidomic studies are supported by transcriptomic analyses (both bulk RNA seq and single cell RNA seq) not performed by the authors but obtained from available public repositories. Bioinformatic analyses confirm that altered glycerophospholipid metabolism is altered in LUAD. I feel that a number of additional questions should be addressed.

1. This study is technically and conceptually similar to a recent report cited by the authors (Wang et al doi: 10.1126/scitranslmed.abk2756) that performed a technically similar approach (LC-MS-based targeted lipidomics assay using MRM model). This manuscript described a 9 lipid diagnostic signature of which only 4 -including the most precise PE(18:0/18:1)- were identified as differentially present by the present study. The authors should discuss the potential causes underlying this discrepancy and the potential implications for the general implementation of their own signature.

Response: Thanks to the reviewer's positive comment. Of the 9 lipid markers (LPC16:0, LPC 18:0, LPC 20:4, PC 16:0-18:1, PC 16:0-18:2, PC 16:0-22:6, PC 18:0-18:1, PC 18:0-18:2, TG 16:0-18:1-18:1) identified by Wang et al, 4 were also detected in the plasma samples in our Discovery cohort, suggesting that their other 5 lipids were not stably detectable in plasma samples of LUADs. In our analysis of searching for lipid markers, their 4 lipid markers were filtered due to their fold changes smaller than 1.5 (or larger than 0.67) between LUADs and healthy people in

the Discovery cohort (PC 16:0-22:6 and PC 18:0-18:1 being up-regulated with fold changes of 1.34 and 1.44, LPC 16:0 and LPC 20:4 being down-regulated with fold change of 0.77 and 0.71 in LUAD compared with healthy people).

We then compared the performance of their 4 lipid markers with that of our 4 lipid markers in the Discovery cohort, finding that their 4 lipid markers performed worse than our 4 lipid signature as shown below or Appendix Figure S8. But, their 4 lipid markers of the 9 lipid diagnostic signature may be used for LUAD diagnosis, which needs to be further validated in independent cohorts. We added this in the Discussion of the manuscript.

ROC curves of signature lipids on the discovery cohort for Wang et al(left) and this study(right).

2. It is unclear for the non-expert to what extent a targeted lipidomic assay would outperform other non-invasive diagnostic approaches such as ctDNA. The authors should comment on this.

Response: Thanks to the reviewer’s positive comment. ctDNA can also be used as cancer biomarkers (Sorenson, Pribish et al., 1994), but has many limitations. Detection sensitivity is a concern as ctDNA levels in early-stage disease are low due to limited tumour DNA shedding and limits on acceptable blood sample volumes, and the low amount of ctDNA may result in a MAF lower than the limit of detection of existing techniques(Garcia-Pardo, Makarem et al., 2022, Wan, Massie et al., 2017).Additionally, ctDNA levels provide a ‘real-time’ snapshot of tumor bulk because of the short half-life of crDNA (2.5 h) (De Rubis, Rajeev Krishnan et al.,

2019), therefore, it need to be separated and extracted as soon as possible after blood sampling, otherwise the false-negative rate of the test may be increased.

The tumor lesion is a complex ecosystem and concentrations of metabolites are highly sensitive to biological activities and pathological conditions, thus, the metabolome may reliably reflect the status of biological systems (Wang, Qiu et al., 2022). We found glycerolipid and glycerophospholipid metabolism pathways down-regulated in tumor cells in early-stage lung cancer tissues compared with epithelial cells in normal lung tissues (Figure 1), consistent with previous observations reported by Wang et al. Moreover, targeted lipidomic assay is a stable method widely used for detecting lipids, which would achieve good performance in early diagnosis of LUAD as we and Wang et al reported. We added this in the Discussion of the manuscript.

3. Similarly, it is not entirely clear how this targeted lipidomic assay would be LUAD specific and not result in cross-diagnostic of other tumour types. As an example, a recent serum lipidomic profiling for pancreatic ductal adenocarcinoma (doi: 10.1038/s41467-021-27765-9) reported a lipid signature that also includes PE(18:0/18:1). This fact should be discussed (and the manuscript cited).

Response: Thanks to the reviewer's positive comment. According to LIPID MAPS database, PE(18:0/18:1) has a synonym named as PE(36:1) which was also identified in pancreatic ductal adenocarcinoma (PDAC) (Wolrab, Jirasko et al., 2022). The levels of PE(36:1) were only identified significantly different between PDAC and normal control in phase II (qualification) with shotgun HR-MS, but not in phase I (discovery) and phase III (verification). It still needs further validation about the performance of PE(36:1) in diagnosing PDAC. Additionally, the levels of PE(36:1) in tumor tissues of PDAC are also needed to be extensively evaluated.

In our LUAD study, the levels of PE (18:0/18:1) was significantly different in both Discovery cohort and plasma validation cohort, which were then confirmed in both tissue validation cohort and cell lines (Figure4A, Figure5E). Altogether, the evidence for PE(18:0/18:1) of cross-diagnostic of other tumour types is not sufficient. We

added this in the Discussion of the manuscript and cited the paper (doi: 10.1038/s41467-021-27765-9).

4. There are increasing evidences suggesting that LUAD displays substantial heterogeneity in energy metabolism within and between tumours (doi: 10.1016/j.cell.2015.12.034). Does the current work suggest that lipid metabolism is more homogeneous? The authors should at least address this question between patients.

Response: Thanks to the reviewer's positive comment. We discovered that lipid metabolism was significantly altered in lung cancer patients compared with the healthy people. The levels of our 4 lipid markers in plasma were found varying among LUADs, demonstrating the characteristics of heterogeneity in regards to lipid metabolism consistent with previous reports(Hensley CT, 2016). But the overall distribution of the levels of our 4 lipid markers among LUADs was significantly different from that in the healthy population (see Figure below, also Appendix Figure S5 A-D). And the similar observation was obtained in the plasma validation cohort (see Figure blow, also Appendix Figure S5 E-H).

The levels of 4 lipid markers in Discovery Cohort. **** represents Pvalue from Mann Whitney U test < 0.0001.

The levels of 4 lipid markers in plasma validation Cohort. **** represents Pvalue from Mann Whitney U test < 0.0001

We also noticed that some LUADs had similar levels of lipid markers to healthy people. To better differentiate LUADs from healthy people, we designed lipid markers-based LSRscore, which was able to increase the difference between LUADs

and healthy people (see Figure below). LSRscore achieved better AUC than each lipid markers in classifying LUADs from healthy people (see Fig3.A and C).

The distribution of LSRscore in the discovery cohort(left) and plasma validation cohort(right).

In short, our results demonstrated that lipid metabolism was heterogeneous among LUADs, but significantly different from that of healthy people.

5. Finally, I am unsure as to whether using publicly available data merits the inclusion of scRNA sequencing as part of the manuscript title.

Response: Thanks to the reviewer's positive comment. Our analysis of scRNA sequencing data that we collected from three sources in the public domain (Appendix Table S1) provided important evidence about the alteration of lipid metabolism in lung cancer, which was the major motivation for us to screen lipid markers for LUAD diagnosis. We would like to include scRNA-seq in the manuscript title. Considering we used publicly available scRNA-seq data, we revised the title. The new manuscript title was revised as 'scRNA-seq data and lipidomics reveals new lipid-based lung adenocarcinoma early diagnosis model'.

Referee #2 (Remarks for Author):

In this manuscript entitled "Single cell RNA-seq and lipidomics reveals new lipid-based lung adenocarcinoma early diagnosis model", Sun et al. propose a lipid signature biomarker for early diagnosis of lung adenocarcinoma. The authors show evidence by scRNAseq of aberrant lipid metabolism associated with lung cancer cells

and identify 4 signature lipid biomarkers by profiling lipidomics in patient samples. Based on this, they develop a plasma lipid signature-based scoring model that they call LSRscore. The authors propose that LSRscore can predict lung adenocarcinoma in a discovery cohort and an independent validation cohort with AUC >0.9. In addition, the authors show that PE(18:0/18:1) lipid biomarker is altered in plasma and tumour tissue thereby being suitable for prediction purposes, and suggest that it is associated with dysregulated glycerophospholipid metabolism.

Overall, the manuscript is interesting, and the developed methodology warrants further investigations due to the unmet clinical need of novel and more refined tools for lung cancer early detection. Such a tool would be of tremendous importance considering the current poor survival rates of lung cancer patients and the significant curable rates when the disease is detected at early stages. However, the existing data do not fully support the claims of the authors in this manuscript and they must clarify some of aspects of their experimental designs and the used databases to validate the efficacy and utility of their lipid signature-based model, see my main points below. The validation of the LSRscore with the independent cohort is questionable. Also, additional mechanistic insights about the interconnection of PE(18:0/18:1) with dysregulated glycerophospholipid metabolism must be provided. Overall, I consider there are major concerns as outlined below that should be addressed to meet with the high quality standards of EMBO Mol Med.

Major concerns:

1. A main point in this study is that both scRNAseq data and RNAseq data have been downloaded from GEO (GSE122960, GSE131907) and TCGA-LUAD/CPTAC-LUAD, but it is unclear the methodology, custom code and raw data used in this study. This information should be clearly provided and deposited in an appropriate repository. Also, it must be clear by referencing in the text (in the Results section) that these are not original data from the authors but collected from databases/publications.

Response: Thanks to the reviewer's positive comment. The methodology used for scRNAseq data and RNAseq had been described in the 'Materials and Methods'

section of the manuscript. The custom code and data used in this study have been uploaded to Biostudies (<https://www.ebi.ac.uk/biostudies/>) with accession number of S-BSST1198.

2. Following my previous comment, the data provided in Figure 1 seem uncomplete and difficult to confirm/corroborate. Marking scores (scales) are missing in the graphs and also the enrichment score of the GSEA (e.g. Fig. 1F).

Response: Thanks to the reviewer's positive comment. We added details for identifying malignant cells from the scRNA-seq data (see Appendix Figure S1) and updated the associated description in the 'Materials and Methods' section of the manuscript. The custom code and data in Figure 1 have been uploaded to Biostudies (<https://www.ebi.ac.uk/biostudies/>) with accession number of S-BSST1198.

GO enrichment result(Fig.1F) was performed using the enrichGO function with a list of differential genes between malignant and non-malignant cells calculated by FindMarkers(qvalue < 0.05 , logFC \geq 0.25). Gosets with qvalue < 0.05 were selected and visualized using Cytoscape and Enrichment Map(Appendix Table S2).The scales were added in Fig.1F.

3. The data presented in the Discovery Cohort make the conclusions speculative and lack of robustness. In particular because of the significant differences between the healthy subjects and LUAD patients in gender and age (i.e. average of 29 years old in the healthy cohort and 59 years old in the LUAD patients). Even for the validation cohort the differences in age (average of 51 years old healthy individuals versus 62 years old LUAD patients) make the conclusions of the authors should be considered carefully. Metabolic and lipidomic profiles can be significantly altered/distinct with age.

Response: Thanks to the reviewer's positive comment. To evaluate whether LSRscore was a measure independent of age, we performed multivariate GLM regression in the discovery cohort and plasma validation cohort respectively, and found that LSRscore was significantly independent of age (p<0.05) (Figure3B,D).

We also analyzed the correlation between the levels of 4 lipid markers and age of LUADs in the Discovery cohort (age range of LUADs: 33 to 80 years old) and plasma validation cohort (age range of LUADs:41 to 83 years old), respectively. We found there were weak correlations between the levels of 4 lipid markers and age (correlation coefficient < 0.2 , Pvalue > 0.05) (see Figure below).

The correlation between the age and the level of lipid for LUADs in the Discovery Cohort.n=60samples.

The correlation between the age and the level of lipid for LUADs in the plasma validation cohort.n=30 samples.

The correlation between the age and the level of lipid for healthy people in the Discovery Cohort.n=30samples.

The correlation between the age and the level of lipid for healthy people in the plasma validation cohort.n=30samples.

Similarly, in the healthy people of the discovery cohort (age range :18 to 38 years old) and plasma validation cohort (age range :47 to 61 years old), the levels of our 4 lipid markers were also weakly correlated with age(correlation coefficient < 0.31 , Pvalue > 0.05 ,see Figure top). These results provided another layer of evidence that

the levels of the 4 lipid markers we identified were not significantly influenced by age.

Additionally, to further evaluate whether there still existed significantly different levels of the 4 lipid markers between age-matched LUAD cohort and healthy people cohort, we extracted all healthy people and LUADs with similar age from the Discovery cohort, obtaining 12 healthy people (mean age:35.25, age range:[33,38]) and 5 LUADs (mean age:39.8, age range: [33,45]) (Welch t-test p-value = 0.12 for age). We discovered that both the levels of the 4 lipid markers and LSRscore were still significantly different between LUADs and healthy people (Figure left below, Appendix Figure S6A). Similar analysis were performed in the plasma validation cohort with 30 healthy people (mean age: 51.57, age range:[47,61]) and 15 LUADs (mean age:54.60, age range:[41,61]) (Welch t-test p-value = 0.1 for age). Similarly, both the levels of the 4 lipid markers and LSRscore were still significantly different between LUADs and healthy people in the plasma validation cohort (Figure right below, Appendix Figure S6B).

Altogether, these results provided multiple layers' evidence that the levels of the 4 lipid markers and LSRscore would not alter with age.

The comparison of lipid concentration between Healthy people and LUADs for lipid markers in the Discovery Cohort(left) and plasma validation cohort(right). The level of 4 lipid markers in Targeted lipidomics data was log2 processed for visualizing. The

P value calculated by Mann Whitney U test.

4. Figure 3C-H. The data of the LSRscore in predicting LUAD with the Independent cohort seem exciting. However, the clinical characteristics of these patients is missing. This information should be provided in the Supplementary Material. For example, are all of these patients treatment naïve? Lipidomic profiles can be altered by treatment conditions. Also, I wonder whether the authors could clarify why the levels of the four lipids in plasma from LUADs and HCs are measured in logarithmic2 scale in Fig 3E-H but not in Fig 2F-I. Checking carefully the information provided in this Figure and considering the scale of the Y axis the differences seem really minimal and hence the validation is questionable.

Response: Thanks to the reviewer’s positive comment. We have added clinical information of LUADs and healthy people in Appendix Table S4-5. All LUADs in this study didn’t receive any treatment at the time of sampling.

Untargeted Lipidomics applied upon the Discovery cohort measured the relative amount of metabolites in each sample, while targeted Lipidomics detected the absolute amount of target metabolite in each sample. As the values of absolute amount of target metabolites were large, for convenience of visualization, we applied log2 to the absolute amount of target metabolites. Here, we replaced Fig 3E-H with the absolute amount of target lipid markers (see Figure below, and Fig 3E-H).

The comparison of lipid concentration in the plasma validation cohort between Healthy people and LUADs for FA(20:0) ,

PE(18:0/18:1), LPC(18:1) and PC(18:2/18:2),. **** represents P value from Mann Whitney U test < 0.0001

5. Figure 5. Coming back to my Point 1, the results shown in Figure 5 need of additional information to be corroborated. All the information regarding clinical

characteristics is missing in the Supplementary Information. Again, the information in the Cancer Genome Atlas is frequently incomplete in terms of these characteristics, for example whether the patients were or not treated and when. This knowledge is very important as per the potential effects of cancer therapies in the glycerophospholipid metabolism.

Response: Thanks to the reviewer’s positive comment. We have added the latest updated clinical data from TCGA-LUAD and CPTAC-LUAD used in Figure5 (Appendix Table S7-8). Treatment information (RadiationTherapy and PharmaceuticalTherapy) was provided in the TCGA-LUAD data. Multivariate Cox regression with treatment information included revealed that the enrichment score for glycerophospholipid metabolism (hsa00564) were associated with overall survival of LUADs, significantly independent of other factors including age, gender, stage, RadiationTherapy and PharmaceuticalTherapy (Figure below, p=0.009, Fig5D). There was no significant difference between the enrichment score and the treatment style(Appendix Figure S4C).

Multivariate Cox regression analysis showed that glycerophospholipid metabolism was associated with overall survival independent of other clinical factors in TCGA-LUAD.

The comparison of enrichment score for glycerophospholipid metabolism(hsa00564) in TCGA-LUAD between LUADs with different treatment. ns represents Pvalue from Mann Whitney U test > 0.1.

6. The authors claim that the overexpression of PE(18:0/18:1) is associated with the dysregulated glycerophospholipid metabolism in the context of LUAD. The results and this hypothesis are stimulating but the authors must provide some mechanistic insights to support this claim to exclude other factors or pathological conditions, such as ageing, inflammation, etc. I find essential to provide a clear evidence of the connection of PE(18:0/18:1) with dysregulated or rewiring glycerophospholipid metabolism for example by using LUAD cell lines. In summary, it is not clear whether upregulated levels of PE(18:0/18:1) are directly or causally connected with LUAD or just indirectly correlated.

Response: Thanks to the reviewer's positive comment. We performed targeted lipidomics assay and RNA sequencing upon cultured LUAD cells(A549) and human normal lung epithelial cells (BEAS-2B) with each having three replicates. The level of PE(18:0/18:1) was significantly higher in LUAD cells than in healthy controls (t-test P=0.046, Figure left below and Fig.5E). We then calculated the enrichment score of glycerophospholipid metabolism using RNA-seq data of A549 and BEAS-2B, finding that glycerophospholipid metabolism had significantly lower enrichment score in A549 cells than in BEAS-2B cells (Figure middle below, Fig.5F, t-test P = 0.03), confirming our discovery that the overexpression of PE(18:0/18:1) is associated with the dysregulated glycerophospholipid metabolism (Figure right below, Fig5G).

Other points/suggestions:

1. English grammar must be revised throughout the text.

Response: Thanks to the reviewer’s positive comment. We carefully revised and polished the text throughout the manuscript.

2. Supplementary Figure 2. Why the training set was done with 70% of the samples and only a 30% of the samples was used for the test set?

Response: Thanks to the reviewer’s positive comment. We split the training and test sets using different ratios (0.6-0.4; 0.7-0.3, 0.8-0.2, 0.9-0.1,1-0; 2000 iterations). The 4 lipid markers were consistently identified at different splitting ratios for training and test sets (see Figure below, Appendix Figure S3B), confirming the robustness of the screening method.

Fig.S3B The overlap of lipid signatures screened by different methods and ratio. Each point represents a lipid on the Y-axis that was screened used method on the X axis. Red dots indicate 4 lipid markers. n = 2000 iterations. RF, Random Forest. . RF6-4: Random Forest applied on training and test sets at ratio of 6:4. RF7-3: Random Forest applied on training and test sets at ratio

of 7:3. RF8-2: Random Forest applied on training and test sets at ratio of 8:2. RF9-1: Random Forest applied on training and test sets at ratio of 9:1. RF10-0: Random Forest applied on training and test sets at ratio of 10:0. Xgboost6-4: XGBOOST applied on training and test sets at ratio of 6:4. Xgboost7-3: XGBOOST applied on training and test sets at ratio of 7:3. Xgboost8-2: XGBOOST applied on training and test sets at ratio of 8:2. Xgboost9-1: XGBOOST applied on training and test sets at ratio of 9:1. Xgboost10-0: XGBOOST applied on training and test sets at ratio of 10:0.

3. Fig. 3E-H. LUADs and HCs are inverted (left and right) in the graph from Figs. 2 and 3, this is confusing. This also happens in Fig. 4 A-C versus Fig. 4B-D.

Response: Thanks to the reviewer's positive comment. We revised the diagrams as suggested.

Referee #3 (Remarks for Author):

The manuscript by Sun et al aimed to develop lipid-based biomarkers for non-invasive diagnosis of early lung adenocarcinoma (LUAD). They found that aberrant lipid metabolisms significantly enriched in lung cancer cells by analyzing single cell RNA-sequencing data from public domain and then identify 4 signature lipid biomarkers by profiling lipidomics of LUADs. They then develop a plasma lipid signature-based scoring model (LSRscore) and that LSRscore has a powerful prediction of LUAD in discovery cohort of 90 subjects (with AUC reaching 0.972) and independent validation cohort of 60 participants (with AUC of 0.924). They demonstrated that a lipid signature biomarker PE(18:0/18:1), is consistently altered in both cancer tissues in situ and plasmas of 25 LUADs, showing strong predictive power for early-stage LUAD. In the transcriptome analysis of 421 LUAD cancer tissues and 120 normal lung tissues show that PE(18:0/18:1) overexpression is associated with dysregulated glycerophospholipid metabolism, which consistently exhibit great prognostic power in two large LUAD cohorts. They suggested that LSRscore and PE(18:0/18:1) may be useful for early-stage diagnosis and survival prediction of LUADs.

The manuscript may be intriguing and the lipid-based biomarkers may be useful if the results can be validated in different institute with larger non-lung cancer cohort.

They are several comments that need to be addressed.

1. The number of normal controls may be too limited, which did not simulated in real world condition; and was not stratified smoking status and other lung conditions, such as COPD, pneumonia and others. it is not known whether the lipid-based biomarkers still workable in inflammatory lung diseases and whether false positive may occur in condition of acute lung injury-repair.

Response: Thanks to the reviewer's positive comment. The ratio of normal controls versus LUADs was 1 vs 2 in the Discovery cohort and 1 vs 1 in plasma validation cohort.

In both Discovery cohort and plasma validation cohort, smoking status, COPD, and bronchiectasis for LUADs, were available for LUADs but not for healthy people. All these clinical information was added in Appendix Table S4 and Appendix Table S5. We discovered that the levels of the 4 lipid markers were not significantly different between LUADs with smoking and LUADs without smoking (see Figure below, Appendix Figure S7A-H).

Lipid marker level corresponding to the different smoking status for LUADs in the discovery cohort(left) and plasma validation cohort(right). p-values were calculated by Student's t-tests

Additionally, there were 3 LUADs with COPD and 1 LUAD with bronchiectasis in the plasma validation cohort. After the removing the 4 LUADs, we consistently found the levels of the 4 lipid markers were significantly altered in LUADs compared with

healthy controls (Figure below, Appendix Figure S7 B-E and Appendix Figure S7 I-L).

The comparison of lipid level in the discovery cohort between healthy people(HC) and LUAD(n=57) without other lung disease for PE(18:0/18:1), PC(18:2/18:2), LPC(18:1) and FA(20:0). **** represents P value from Mann Whitney U test < 0.0001

The comparison of lipid level in the plasma validation cohort between healthy people(HC) and LUAD(n=26) without other lung disease for PE(18:0/18:1), PC(18:2/18:2), LPC(18:1) and FA(20:0). **** represents P value from Mann Whitney U test < 0.0001

2. Although the authors claimed that the LSRscore may be helpful in clinical applications to improve clinical detection and screening for LUAD without invasive diagnostic procedures or unnecessary exposure to radiation as required by LDCT. If they can demonstrate that combination of lipid-based assay and LDCT can further improve the diagnostic accuracy, the study will be even more powerful.

Response: Thanks to the reviewer's positive comment. In both Discovery cohort and plasma validation cohort, all LUAD patients exhibited aberration according to LDCT and pathologically confirmed as LUAD. Also, LSRscore successfully classified LUADs from healthy people. The combination of LDCT and lipid-based assay would improve the diagnostic accuracy, which needs to be further evaluated in more data. In the future, we would collect more data from healthy people with aberrant phenotype in LDCT and early stage LUAD patients with normal phenotype based on LDCT.

3. *In line 6 of introduction, the citation of 5-year survival should be better include both overall and stage 1. The citation of stage 1 survival needs to be updated to the most recent one. The 5-year survival of 60% in stage 1 disease seems not meet the current standard of care.*

Response: Thanks to the reviewer's positive comment. We have modified the text accordingly. The five-year overall survival (OS) rate of LUAD across all stages is approximately 18% (Fang, Sun et al., 2023), and the 5-year recurrence-free survival rate of stage IA ranges from 63% to 81%(Carr, Wang et al., 2023).

4. *In the result first paragraph, "To discover reliable biomarkers for early-stage lung adenocarcinoma (LUAD) diagnosis, we collected single cell RNA sequencing (scRNA-seq) data of tumor and adjacent normal tissues from 17 LUADs and eight healthy lung samples from healthy donors in the public domain." The source of public domain should be cited. The demographic information including age, gender, smoking status, stage and driver gene status should be described. How these 17 LUAD were selected?*

Response: Thanks to the reviewer's positive comment. We added clinical information of single-cell RNA data from public sources in Appendix Table S1, and the corresponding data and codes have been uploaded to Biostudies (<https://www.ebi.ac.uk/biostudies/>) with accession number of S-BSST1198.

Of all scRNA-seq data, there were a total of 17 LUADs with TNM stage I and II. In this study, we focused on early stage LUAD, therefore, we selected these 17 LUADs.

5. *In supplementary table 1 which summarized the demographic information of study population. The readers will be interested to know in more detail the smoking status and underlying diseases in normal subjects. It is also important to know if the lipid profile altered in different driver gene mutations, such as EGFR and Kras, especially EGFR which is nearly 50% in LUAD in China, it may be important to know whether the lipid-based biomarkers can be applied independent of all different drivers.*

Response: Thanks to the reviewer's positive comment. We added mutation information of EGFR in Appendix Table S4-5.

In the Discovery cohort, a total of 15 LUADs were tested for mutations of EGFR genes. Of 15 LUADs, 10 patients had EGFR mutations (Appendix Table S4). We discovered that the levels of the 4 lipid markers and LSRscore did not exhibit significant changes between LUADs with EGFR mutations and LUADs without EGFR mutations (see Figure below and Appendix Figure S7F-G).

Similarly, in the plasma validation cohort, a total of 9 LUADs were tested for mutations of EGFR genes. Of 9 LUADs, 5 patients had EGFR mutations (Appendix Table S5). We discovered that the levels of the 4 lipid markers did not exhibit significant changes between LUADs with EGFR mutations and LUADs without EGFR mutations (see Figure below and Appendix Figure S7 M-N).

Lipid levels and LSRscore levels corresponding to the different mutation status of sample EGFR in the discovery cohort. p-values were calculated by Student's t-tests.

Lipid levels and LSRscore levels corresponding to the different mutation status of sample EGFR in the plasma validation cohort.

p-values were calculated by Student's t-tests.

Additionally, in the tissue validation cohort, a total of 8 LUAD patients were tested for EGFR mutations. 4 of 8 LUADs contained EGFR mutations. We found the levels of PE(18:0/18:1) did not display significant changes between LUADs with EGFR mutations and LUADs without EGFR mutations (Figure below, Appendix Figure S7 O)

PE(18:0/18:1) levels in tumor tissue corresponding to the different mutation samples in the tissue validation cohort. p-values were calculated by Student's t-tests.

References:

Carr SR, Wang H, Hudlikar R, Lu X, Zhang MR, Hoang CD, Yan F, Schrupp DS (2023) A Unique Gene Signature Predicting Recurrence Free Survival in Stage IA Lung Adenocarcinoma. *J Thorac Cardiovasc Surg* 165: 1554-1564

De Rubis G, Rajeev Krishnan S, Bebawy M (2019) Liquid Biopsies in Cancer Diagnosis, Monitoring, and Prognosis. *Trends Pharmacol Sci* 40: 172-186

Fang H, Sun Q, Zhou J, Zhang H, Song Q, Zhang H, Yu G, Guo Y, Huang C, Mou Y, Jia C, Song Y, Liu A, Song K, Lu C, Tian R, Wei S, Yang D, Chen Y, Li T et al. (2023) m(6)A methylation reader IGF2BP2 activates endothelial cells to promote angiogenesis and metastasis of lung adenocarcinoma. *Mol Cancer* 22: 99

Garcia-Pardo M, Makarem M, Li JJN, Kelly D, Leighl NB (2022) Integrating circulating-free DNA (cfDNA) analysis into clinical practice: opportunities and challenges. *Br J Cancer* 127: 592-602

Hensley CT FB, Yuan Q, Lev-Cohain N, Jin E, Kim J, Jiang L, Ko B, Skelton R, Loudat L, Wodzak M, Klimko C, McMillan E, Butt Y, Ni M, Oliver D, Torrealba J, Malloy CR, Kernstine K, Lenkinski RE, DeBerardinis RJ (2016) Metabolic Heterogeneity in Human Lung Tumors *Cell* 164: 681-94

Sorenson GD, Pribish DM, Valone FH, Memoli VA, Bzik DJ, Yao SL (1994) Soluble normal and mutated DNA sequences from single-copy genes in human blood. *Cancer Epidemiol Biomarkers Prev* 3: 67-71

Wan JCM, Massie C, Garcia-Corbacho J, Mouliere F, Brenton JD, Caldas C, Pacey S, Baird R, Rosenfeld N (2017) Liquid biopsies come of age: towards implementation of circulating tumour DNA. *Nat Rev Cancer* 17: 223-238

Wang G, Qiu M, Xing X, Zhou J, Yao H, Li M, Yin R, Hou Y, Li Y, Pan S, Huang Y, Yang F, Bai F, Nie H, Di S, Guo L, Meng Z, Wang J, Yin Y (2022) Lung cancer scRNA-seq and lipidomics reveal aberrant lipid metabolism for early-stage diagnosis. *Sci Transl Med* 14: eabk2756

Wolrab D, Jirasko R, Cifkova E, Horing M, Mei D, Chocholouskova M, Peterka O, Idkowiak J, Hrciarova T, Kuchar L, Ahrends R, Brumarova R, Friedecky D, Vivo-Truyols G, Skrha P, Skrha J, Kucera R, Melichar B, Liebisch G, Burkhardt R et al. (2022) Lipidomic profiling of human serum enables detection of pancreatic cancer. *Nat Commun* 13: 124

26th Oct 2023

Dear Dr. Zhang,

Thank you for the submission of your revised manuscript to EMBO Molecular Medicine. We have now heard back from the three referees who we asked to re-evaluate your manuscript. As you will see from the reports below, while referees #1 and #3 support publication of the manuscript, referee #2 acknowledges the improvements of the revised manuscript but remains critical particularly regarding, but not limited to, the unclear association of PE(18:0/18:1) with dysregulated glycerophospholipid metabolism in the context of LUAD and the limited conceptual advance. During our cross-commenting discussion both referee #1 and #3 agreed that the concerns raised by the referee #2 are justified and should be addressed in an additional and final round of major revision. Furthermore, we would like to ask you to revise the title taking in consideration the referee #1 initial comment reiterated by the referee #2 that reanalyzed scRNA sequencing data should not be part of the title, as, also in our opinion, it does not accurately reflect the experimental work.

Further consideration of a revision that addresses reviewer's concerns in full will entail an additional round of review. Acceptance or rejection of the manuscript will depend on the completeness of your responses included in the next, final version of the manuscript. For this reason, and to save you from any frustrations in the end, I would strongly advise against returning an incomplete revision.

We would welcome the submission of a revised version within three months for further consideration. Please let us know if you require longer to complete the revision.

I look forward to seeing a revised form of your manuscript as soon as possible.

I look forward to receiving your revised manuscript.

Yours sincerely,

Zeljko Durdevic

We require:

2) Individual production quality figure files as .eps, .tif, .jpg (one file per figure). For guidance, download the 'Figure Guide PDF': (<https://www.embopress.org/page/journal/17574684/authorguide#figureformat>).

3) A .docx formatted letter INCLUDING the reviewers' reports and your detailed point-by-point responses to their comments. As part of the EMBO Press transparent editorial process, the point-by-point response is part of the Review Process File (RPF), which will be published alongside your paper.

4) A complete author checklist, which you can download from our author guidelines (<https://www.embopress.org/page/journal/17574684/authorguide#submissionofrevisions>). Please insert information in the checklist that is also reflected in the manuscript. The completed author checklist will also be part of the RPF.

6) It is mandatory to include a 'Data Availability' section after the Materials and Methods. Before submitting your revision, primary datasets produced in this study need to be deposited in an appropriate public database, and the accession numbers and database listed under 'Data Availability'. Please remember to provide a reviewer password if the datasets are not yet public (see <https://www.embopress.org/page/journal/17574684/authorguide#dataavailability>).

13) Author contributions: You will be asked to provide CRediT (Contributor Role Taxonomy) terms in the submission system.

These replace a narrative author contribution section in the manuscript.

14) A Conflict of Interest statement should be provided in the main text.

Please note: When submitting your revision you will be prompted to enter your funding and payment information. This will allow Wiley to send you a quote for the article processing charge (APC) in case of acceptance. This quote takes into account any reduction or fee waivers that you may be eligible for. Authors do not need to pay any fees before their manuscript is accepted and transferred to the publisher.

EMBO Press participates in many Publish and Read agreements that allow authors to publish Open Access with reduced/no publication charges. Check your eligibility: <https://authorservices.wiley.com/author-resources/Journal-Authors/open-access/affiliation-policies-payments/index.html>

***** Reviewer's comments *****

Referee #1 (Remarks for Author):

The authors have convincingly addressed all my requests.

Referee #2 (Remarks for Author):

This is now the revised version of the manuscript entitled "Single cell RNA-seq and lipidomics reveals new lipid-based lung adenocarcinoma early diagnosis model" by Sun et al. providing a lipid signature biomarker for early diagnosis of lung adenocarcinoma. Although the authors have addressed some of my concerns and those of the other Reviewers I still consider additional work must be done to meet with high quality standards of EMBO Mol Med. At this stage, the publication is not still warranted due to the reasons stated below, in particular:

Major concerns

1. I agree with Referee 1 about the merits of using publicly available data to include scRNA sequencing as part of the manuscript title. The main conclusion of the first Results section is that lipid metabolism dysregulation occurs in early stage lung cancer and can be used to identify LUAD tumour cells. But this conclusion is not demonstrated by the scRNAseq analysis of the authors, which is supported by Fig. 1F and Appendix Table S2, but majorly stands in the recent paper by Wang G et al (Sci Transl Med 2022), which indeed is entitled "Lung cancer scRNA-seq and lipidomics reveal aberrant lipid metabolism for early-stage diagnosis". Even more importantly, a similar 9 lipid diagnostic signature is already described in this paper. Altogether, the results in the Sci Transl Med work questions the originality of the submitted manuscript.

2. Considering the available data I still think it is unclear whether upregulated levels of PE(18:0/18:1) are causally connected or just indirectly correlated with the rewiring glycerophospholipid metabolism. The authors provide some incomplete data in new Figures 5E-G by comparing PE(18:0/18:1) levels by targeted lipidomics in A549 and BEAS-2B cells. Although apparently statistically significant, the observed differences as per the scale of the y-axis (log₂ 11.6-12.3) are really minimal. Also, in terms of robustness, this experiment should be done by using several LUAD/control cell lines, including other cancer cell types (distinct from the lung) as a negative control (see the potential cross-diagnostic issue of lipidomics in other cancer types from Reviewer 1).

3. In addition, the RNAseq data shown in Fig. 5F-G are largely incomplete. The authors should provide GSEA of relevant genes together with the enrichment pathway analyses. Crucially, functional evidence of the dysregulation of the glycerophospholipid metabolism should be provided by exposing cells to increasing concentrations of PE(18:0/18:1), which is available by different Companies.

Overall, additional mechanistic insights are required to support the claim that the overexpression of PE(18:0/18:1) is associated with the dysregulated glycerophospholipid metabolism in the context of LUAD.

Other points

1. Why the UMAPS of Figure 1 are shown in a different manner than in the previous submission? It seems that the images have been rotated or inverted.
2. For consistency, the comparisons of lipid concentration in Fig. 2F-I, Fig. 3E-H and Fig. 5E should be standardized in terms of the notation of the y axis (decimal numeral, log2, e notation, etc.).
3. Figure 5B seems uncomplete in the current submission when compared to the initial submission. Also, the p-value has shifted from 0.015 to 0.00014, as well as the 0.5 survival probability line, while it is essentially the same study and numbers of patients. How is that?
4. Overall, the claims of the authors should be toned down and the limitations of the study acknowledged.

Referee #3 (Remarks for Author):

I have no further comments.

******* Reviewer's comments *********Referee #1 (Remarks for Author):**

The authors have convincingly addressed all my requests.

Response: Thanks for the positive comments of the reviewer.

Referee #2 (Remarks for Author):

This is now the revised version of the manuscript entitled "Single cell RNA-seq and lipidomics reveals new lipid-based lung adenocarcinoma early diagnosis model" by Sun et al. providing a lipid signature biomarker for early diagnosis of lung adenocarcinoma. Although the authors have addressed some of my concerns and those of the other Reviewers I still consider additional work must be done to meet with high quality standards of EMBO Mol Med. At this stage, the publication is not still warranted due to the reasons stated below, in particular:

Major concerns

1. I agree with Referee 1 about the merits of using publicly available data to include scRNA sequencing as part of the manuscript title. The main conclusion of the first Results section is that lipid metabolism dysregulation occurs in early stage lung cancer and can be used to identify LUAD tumour cells. But this conclusion is not demonstrated by the scRNAseq analysis of the authors, which is supported by Fig. 1F and Appendix Table S2, but majorly stands in the recent paper by Wang G et al (Sci Transl Med 2022), which indeed is entitled "Lung cancer scRNA-seq and lipidomics reveal aberrant lipid metabolism for early-stage diagnosis". Even more importantly, a similar 9 lipid diagnostic signature is already described in this paper. Altogether, the results in the Sci Transl Med work questions the originality of the submitted manuscript.

Response: Thanks for the positive comments of the reviewer. Our analysis of publicly available scRNA sequencing data (Appendix Table S1) identified the alteration of lipid metabolism in lung cancer. Considering we used publicly available scRNA-seq data, we revised the title. The new manuscript title was revised as '*Lipidomics reveal aberrant lipid metabolism for early-stage diagnosis*'.

2. Considering the available data I still think it is unclear whether upregulated levels of PE(18:0/18:1) are causally connected or just indirectly correlated with the rewiring glycerophospholipid metabolism. The authors provide some incomplete data in new Figures 5E-G by comparing PE(18:0/18:1) levels by targeted lipidomics in A549 and BEAS-2B cells. Although apparently statistically significant, the observed differences as per the scale of the y-axis (\log_2 11.6-12.3) are really minimal. Also, in terms of robustness, this experiment should be done by using several LUAD/control cell lines, including other cancer cell types (distinct from the lung) as a negative control (see the potential cross-diagnostic issue of lipidomics in other cancer types from Reviewer 1).

Response: Thanks for the positive comments of the reviewer. Targeted lipidomics were performed for cultured LUAD cells (A549) and human normal lung epithelial cells (BEAS-2B) with each having three replicates (Fig. 5E, Appendix Table S10). The PE(18:0/18:1) levels of LUAD cells (A549) were significantly higher, ranging from 4050.92 to 5061.40, while BEAS-2B cells ranged from 3301.97 to 3931.73 and we visualized the expression levels \log_2 processed (Figures 5E-G).

To further validate that PE(18:0/18:1) is specifically highly expressed in LUAD, we applied targeted lipidomics analysis upon two LUAD cell lines (A549 and NCI-H1975), and three control cell lines including two normal human cell lines (human normal lung epithelial cells, BEAS-2B; human skin fibroblasts, HSF) and one human liver carcinoma cell line (HepG2), with each having three replicates (Appendix Table S6). We discovered that the levels of PE(18:0/18:1) were significantly higher in LUAD cells than that in normal control and liver carcinoma cell lines (t-test, $P=0.004$ for A549 and BEAS-2B; $P=0.05$ for A549 and HSF; $P=0.01$ for NCI-H1975 and BEAS-2B; $P=0.026$ for NCI-H1975 and HSF; $P=0.015$ for A549 and HepG2; $P=0.02$ for NCI-H1975 and HepG2) (Fig. 4G, or Figure below). The level of PE(18:0/18:1) in liver carcinoma cell line was not significantly higher than those in normal cell lines (t-test, $P=0.13$ for HepG2 and BEAS-2B; $P=0.2$ for HepG2 and HSF) (Fig.4G, or Figure below). There were no significant difference either between two LUAD cell lines (t-test, $P=0.34$ for

A549 and NCI-H1975) or between two normal cell lines (t-test, $P=0.46$ for BEAS-2B and HSF) (Fig.4G or Figure below). These results from in vitro confirmed that PE(18:0/18:1) is specifically upregulated in LUAD.

To verify whether the levels of PE(18:0/18:1) may directly affect the dysregulation of glycerophospholipid metabolism, we treated BEAS-2B cell lines with PE(18:0/18:1) applied at different concentrations (0 $\mu\text{mol/L}$, 1 $\mu\text{mol/L}$, 10 $\mu\text{mol/L}$, and 100 $\mu\text{mol/L}$), followed by RNA sequencing (each having three replicates). GO enrichment analysis demonstrated that there were no significant enriched GO terms between BEAS-2B cell lines treated with PE(18:0/18:1) at 1 $\mu\text{mol/L}$ and those at 0 $\mu\text{mol/L}$ (as control). When the PE(18:0/18:1) concentration increased to 10 $\mu\text{mol/L}$, we observed that some lipid metabolism related pathways were down regulated in BEAS-2B cell lines (Appendix Figure 5A-B, or Figure below A-B). Particularly, a great number of lipid metabolism related pathways including glycerophospholipids metabolism were down-regulated in BEAS-2B cell lines treated with PE(18:0/18:1) at 100 $\mu\text{mol/L}$ (Appendix Figure 5C-E or Figure below C-E). There was a clear trend that the increasing concentration of PE(18:0/18:1) was associated with the decreasing enrichment score of glycerophospholipid metabolism (Appendix Figure S5F, or Figure below F).

A. Compared with the control(0 $\mu\text{mol/L}$), the pathways down-regulated in PE 10 $\mu\text{mol/L}$. B. Compared with the PE 1 $\mu\text{mol/L}$, the pathways down-regulated in PE 10 $\mu\text{mol/L}$. C. Compared with the control, the pathways down-regulated in PE 100 $\mu\text{mol/L}$. D. Compared with the PE 10 $\mu\text{mol/L}$, the pathways down-regulated in PE 100 $\mu\text{mol/L}$. E. Compared with the PE 1 $\mu\text{mol/L}$, the pathways down-regulated in PE 100 $\mu\text{mol/L}$. F. The enrichment status of glycerophospholipid metabolism pathways with different concentrations PE(18:0/18:1). with 3 independent repetitions. Red arrows indicate of glycerophospholipid metabolism pathway.

3. In addition, the RNAseq data shown in Fig. 5F-G are largely incomplete. The authors should provide GSEA of relevant genes together with the enrichment pathway analyses. Crucially, functional evidence of the dysregulation of the glycerophospholipid metabolism should be provided by exposing cells to increasing concentrations of PE(18:0/18:1), which is available by different Companies. Overall, additional mechanistic insights are required to support the claim that the overexpression of PE(18:0/18:1) is associated with the dysregulated glycerophospholipid metabolism in the context of LUAD.

Response: Thanks to the reviewer's positive comment. Targeted lipidomics assay and RNA sequencing upon cultured LUAD cells(A549) and human normal lung epithelial cells (BEAS-2B) with each having three replicates. We calculated the enrichment score of glycerophospholipid metabolism using RNA-seq data of A549 and BEAS-2B, finding that glycerophospholipid metabolism had significantly lower enrichment score in A549 cells than in BEAS-2B cells (t-test, $P=0.03$, Fig.5F, Figure left), indicating the

overexpression of PE(18:0/18:1) is associated with the dysregulated glycerophospholipid metabolism. Next, Differential genes were selected (FDRpvalue <0.05 ,FoldChange < 0.67 A549 down-regulated) and GO enrichment analysis was performed, finding significant down-regulation of pathways associated with lipid metabolism (including glycerophospholipid metabolism) (Figure right).

To further validate whether the level of PE(18:0/18:1) affects the dysregulation of glycerophospholipid metabolism, BEAS-2B cells were cultured with different concentrations of PE(18:0/18:1) (0 μ mol/L, 1 μ mol/L, 10 μ mol/L, 100 μ mol/L). GO enrichment analysis showed lipid metabolic related pathways down-regulation as PE(18:0/18:1) concentrations increased (Appendix Figure S5 or Figure below). The glycerophospholipids metabolism was down regulated when the PE(18:0/18:1) concentration was 100 μ mol/L (Appendix Figure S5 C-E or Figure below C-E). Additionally, the enrichment score of glycerophospholipid metabolism indicating a decreasing trend with increasing PE concentrations (Appendix Figure S5F, Figure below F).

A. Compared with the control(0 $\mu\text{mol/L}$), the pathways down-regulated in PE 10 $\mu\text{mol/L}$. B. Compared with the PE 1 $\mu\text{mol/L}$, the pathways down-regulated in PE 10 $\mu\text{mol/L}$. C. Compared with the control, the pathways down-regulated in PE 100 $\mu\text{mol/L}$. D. Compared with the PE 10 $\mu\text{mol/L}$, the pathways down-regulated in PE 100 $\mu\text{mol/L}$. E. Compared with the PE 1 $\mu\text{mol/L}$, the pathways down-regulated in PE 100 $\mu\text{mol/L}$. F. The enrichment status of glycerophospholipid metabolism pathways with different concentrations PE(18:0/18:1). with 3 independent repetitions. Red arrows indicate of glycerophospholipid metabolism pathway.

Other points

1. Why the UMAPS of Figure 1 are shown in a different manner than in the previous submission? It seems that the images have been rotated or inverted.

Response: Thanks to the reviewer's positive comment. The rotation of the UMAP image was caused by the inconsistency by the two seurat versions and no seed was set at runtime. Although the first version is 4.1.0 and the second version is 4.1.3, the clustering results are the same for both units.

2. For consistency, the comparisons of lipid concentration in Fig. 2F-I, Fig. 3E-H and Fig. 5E should be standardized in terms of the notation of the y axis (decimal numeral, log₂, e notation, etc.).

Response: Thanks to the reviewer's positive comment. We have modified the corresponding images to visualize the lipid concentrations after log₂ processing.

3. Figure 5B seems uncomplete in the current submission when compared to the initial submission. Also, the p-value has shifted from 0.015 to 0.00014, as well as the 0.5 survival probability line, while it is essentially the same study and numbers of patients. How is that?

Response: Thanks to the reviewer's positive comment. In current submission, CPTAC-LUAD transcriptome profiling data was re-downloaded (2023-09), and compared with the clinical data originally submitted (2022-08), we found the survival data of some samples was updated (Table 1). Among them, the survival state of 7 samples had changed, and the last follow up data of 34 alive samples had changed.

Next, GSEA was applied to evaluate the enrichment status of each pathways incorporated PE(18:0/18:1) for each patient in CPTAC-LUAD in the initial submission, while normalized enrichment score (NES)(Zhang, Caruso et al., 2019) was used to evaluate the changes enrichment of each metabolic pathway related(see Materials and Methods) in current submission. We compared the two assessment methods simultaneously, and the results are shown in the figure below:

A,B:Kaplan-Meier curves show the overall survival of LUAD patients with high(>median) or low(\leq median) enrichment of glycerophospholipid metabolism in CPTAC-LUAD using Normalized Enrichment Score for A(2022year) and B(2023year).

C,D:Kaplan-Meier curves show the overall survival of LUAD patients with high(>median) or low(\leq median) enrichment of glycerophospholipid metabolism in CPTAC-LUAD using GSEA for C(2022year) and D(2023year).

Table1 : Samples with changes in clinical informations in CPTAC-LUAD.

case_submitter_id	vital_status_2022	days_to_death_2022	days_to_last_follow_up_2022	vital_status_2023	days_to_death_2023	days_to_last_follow_up_2023
C3N-03765	Alive	..	348	Dead	827	719
C3L-04757	Alive	..	686	Dead	1135	1010
C3N-02281	Alive	..	787	Dead	1436	1436
C3N-04176	Alive	..	-27	Dead	289	289
C3N-03420	Alive	..	455	Dead	1135	1130
C3N-04180	Alive	..	344	Dead	709	344
C3N-02950	Alive	..	1219	Dead	1359	1219
C3N-02672	Alive	..	730	Alive	..	1425
C3L-03679	Alive	91	1007	Alive	91	1343
C3N-02451	Alive	..	1386	Alive	..	1735
C3L-02513	Alive	..	601	Alive	..	1336
C3L-02350	Alive	..	1418	Alive	..	1950
C3L-02961	Alive	..	1064	Alive	..	1424
C3L-02967	Alive	..	704	Alive	..	1519
C3N-02282	Alive	..	737	Alive	..	1831
C3L-02958	Alive	..	1044	Alive	..	1432
C3L-02345	Alive	..	1376	Alive	..	1796
C3L-01330	Alive	..	1317	Alive	..	1595
C3L-01889	Alive	..	1355	Alive	..	1799
C3L-02515	Alive	..	1176	Alive	..	1818
C3L-00422	Alive	..	1426	Alive	..	1916
C3N-01799	Alive	..	1412	Alive	..	1545
C3L-03462	Alive	..	1051	Alive	..	1436
C3L-03721	Alive	..	1306	Alive	..	1341
C3L-02559	Alive	..	1421	Alive	..	1764
C3L-03463	Alive	..	1013	Alive	..	1373
C3L-00444	Alive	..	1429	Alive	..	1919
C3L-02348	Alive	..	1425	Alive	..	1791
C3N-02973	Alive	..	1475	Alive	..	1548
C3L-04033	Alive	..	993	Alive	..	1405
C3N-00959	Alive	..	1519	Alive	..	1758
C3L-00973	Alive	..	1131	Alive	..	1435
C3N-00294	Alive	..	1042	Alive	..	1848
C3L-00604	Alive	..	1374	Alive	..	1801
C3L-02169	Alive	..	1140	Alive	..	1475
C3L-03726	Alive	..	749	Alive	..	1365
C3L-01682	Alive	..	1617	Alive	..	1799
C3L-03985	Alive	..	989	Alive	..	1367
C3N-00293	Alive	..	14	Alive	..	1832
C3N-04173	Alive	..	355	Alive	..	1501
C3L-02560	Alive	..	1427	Alive	..	1810

4. Overall, the claims of the authors should be toned down and the limitations of the study acknowledged.

Response: Thanks to the reviewer's positive comment. We've added the limitations in the discussion section.

Referee #3 (Remarks for Author):

I have no further comments.

Response: Thanks for the positive comments of the reviewer.

References:

Zhang J, Caruso FP, Sa JK, Justesen S, Nam DH, Sims P, Ceccarelli M, Lasorella A, Iavarone A (2019)
The combination of neoantigen quality and T lymphocyte infiltrates identifies glioblastomas with
the longest survival. Commun Biol 2: 135.

13th Feb 2024

Dear Dr. Zhang,

Thank you for the submission of your revised manuscript to EMBO Molecular Medicine. I am pleased to inform you that we will be able to accept your manuscript pending the following final amendments:

- 1) Authors: Please provide institutional e-mail addresses for Jing Zhang and Xun Wang.
- 2) In the main manuscript file, please do the following:
 - Please address all comments suggested by our data editors listed below:
 - o Figure legends:
 1. Please note that legend for figure 2j should start on a new line OR it should be labelled as 2F-J.
 2. Please indicate the statistical test used for data analysis in the legend of figure 5b
 3. Please note that the box plots need to be defined in terms of minima, maxima, centre, bounds of box and whiskers, and percentile in the legend of figures 2f-i; 3e-h; 4a, c.
 4. Please note that the error bars are not defined in the legend of figures 5c-d, e-f
 - Please move the ethics statement to M&M "Patient enrollment" paragraph and confirm that in addition to the principals of the Declaration of Helsinki the experiments also conformed to the principles set out in the Department of Health and Human Services Belmont Report.
 - Remove consent for publication statement.
 - Author contributions: Please remove it from the manuscript and specify author contributions in our submission system. CRediT has replaced the traditional author contributions section because it offers a systematic machine-readable author contributions format that allows for more effective research assessment. You are encouraged to use the free text boxes beneath each contributing author's name to add specific details on the author's contribution. More information is available in our guide to authors:
<https://www.embopress.org/page/journal/17574684/authorguide#authorshipguidelines>
 - In data availability statement please provide resolvable links to deposited data. Use the following format to report the accession number of your data:

[data type]: [full name of the resource] [accession number/identifier] [(doi or URL or identifiers.org/DATABASE:ACCESSION)]

Please check "Author Guidelines" for more information.

<https://www.embopress.org/page/journal/17574684/authorguide#availabilityofpublishedmaterial>

- Correct the reference citation in the reference list. Where there are more than 10 authors on a paper, 10 will be listed, followed by "et al.". Please check "Author Guidelines" for more information.

<https://www.embopress.org/page/journal/17574684/authorguide#referencesformat>

3) Appendix: Please add page numbers to the table of content.

4) Funding: Please merge it with "Acknowledgments".

5) Synopsis:

- Synopsis image: Please upload the image as a separate, high-resolution jpeg file 550 px-wide x (250-400)-px high.

6) As part of the EMBO Publications transparent editorial process initiative (see our Editorial at

<http://embomolmed.embopress.org/content/2/9/329>), EMBO Molecular Medicine will publish online a Review Process File (RPF) to accompany accepted manuscripts. This file will be published in conjunction with your paper and will include the anonymous referee reports, your point-by-point response and all pertinent correspondence relating to the manuscript. Let us know whether you agree with the publication of the RPF and as here, if you want to remove or not any figures from it prior to publication.

7) Please provide a point-by-point letter INCLUDING my comments as well as the reviewer's reports and your detailed responses (as Word file).

I look forward to reading a new revised version of your manuscript as soon as possible.

Yours sincerely,

Zeljko Durdevic

Zeljko Durdevic
Editor

*** Instructions to submit your revised manuscript ***

- 1) a .docx formatted version of the manuscript text (including Figure legends and tables)
- 2) Separate figure files*
- 3) supplemental information as Expanded View and/or Appendix. Please carefully check the authors guidelines for formatting Expanded view and Appendix figures and tables at <https://www.embopress.org/page/journal/17574684/authorguide#expandedview>
- 4) a letter INCLUDING the reviewer's reports and your detailed responses to their comments (as Word file).
- 5) The paper explained: EMBO Molecular Medicine articles are accompanied by a summary of the articles to emphasize the major findings in the paper and their medical implications for the non-specialist reader. Please provide a draft summary of your article highlighting
 - the medical issue you are addressing,
 - the results obtained and
 - their clinical impact.This may be edited to ensure that readers understand the significance and context of the research. Please refer to any of our published articles for an example.
- 6) For more information: There is space at the end of each article to list relevant web links for further consultation by our readers. Could you identify some relevant ones and provide such information as well? Some examples are patient associations, relevant databases, OMIM/proteins/genes links, author's websites, etc...
- 7) Author contributions: the contribution of every author must be detailed in a separate section.
- 8) EMBO Molecular Medicine now requires a complete author checklist (<https://www.embopress.org/page/journal/17574684/authorguide>) to be submitted with all revised manuscripts. Please use the checklist as guideline for the sort of information we need WITHIN the manuscript. The checklist should only be filled with page numbers where the information can be found. This is particularly important for animal reporting, antibody dilutions (missing) and exact values and n that should be indicated instead of a range.
- 9) Every published paper now includes a 'Synopsis' to further enhance discoverability. Synopses are displayed on the journal webpage and are freely accessible to all readers. They include a short stand first (maximum of 300 characters, including space) as well as 2-5 one sentence bullet points that summarise the paper. Please write the bullet points to summarise the key NEW findings. They should be designed to be complementary to the abstract - i.e. not repeat the same text. We encourage inclusion of key acronyms and quantitative information (maximum of 30 words / bullet point). Please use the passive voice. Please attach these in a separate file or send them by email, we will incorporate them accordingly.

You are also welcome to suggest a striking image or visual abstract to illustrate your article. If you do please provide a jpeg file 550 px-wide x 300-800px high.

10) A Conflict of Interest statement should be provided in the main text

11) Please note that we now mandate that all corresponding authors list an ORCID digital identifier. This takes <90 seconds to complete. We encourage all authors to supply an ORCID identifier, which will be linked to their name for unambiguous name identification.

Currently, our records indicate that the ORCID for your account is 0000-0001-8549-3286.

Link Not Available

Photos 400-800 DPI

*Additional important information regarding figures and illustrations can be found at

<https://bit.ly/EMBOPressFigurePreparationGuideline>. See also figure legend preparation guidelines:

<https://www.embopress.org/page/journal/17574684/authorguide#figureformat>

***** Reviewer's comments *****

Referee #2 (Comments on Novelty/Model System for Author):

The authors have addressed now my concerns and I have no further requests or comments.

Referee #2 (Remarks for Author):

I have no further requests or comments. I congratulate the authors for the work done.

The authors addressed the remaining editorial issues.

27th Feb 2024

Dear Dr. Zhang,

We are pleased to inform you that your manuscript is accepted for publication and is now being sent to our publisher to be included in the next available issue of EMBO Molecular Medicine.
